# Utilizing Label Smoothing as a Tool for Embedding Perturbation Uncertainty

## Abstract

Model robustness is the ability of a machine learning model to perform well when confronted with unexpected distributional shifts during inference. While various augmentation-based methods exist to improve common corruption robustness, they often rely on predefined image operations, and the untapped potential of perturbation-based strategies still exists. In response to these limitations, we repurpose label smoothing as a tool for embedding the uncertainty of perturbations. By correlating confidence levels with a monotonically decreasing function to the intensity of isotropic perturbations, we demonstrate that the model eventually acquires the increased boundary thickness and flatter minima. These metrics have strong relationships with general model robustness, extending beyond the resistance to common corruption. Our evaluations on CIFAR-10/100, Tiny-ImageNet, and ImageNet benchmarks confirm that our approach not only bolsters robustness on its own but also complements existing augmentation strategies effectively. Notably, our method enhances both common corruption and adversarial robustness in all experimental cases, a feature not observed with prior augmentations.

## 1 Introduction

Model robustness, or its ability to maintain performance when data distribution unexpectedly shifts at inference time, is a critical aspect of machine learning. Despite their importance in high-stakes fields like autonomous driving and medical diagnostics, Deep Neural Networks often struggle in the face of minor but humanly imperceptible data perturbations (Szegedy et al., 2013; Goodfellow et al., 2015).

Robustness varies based on distributional shift, whether it be common corruption (Hendrycks et al., 2022; He et al., 2021), adversarial corruption (Xu et al., 2023; Goodfellow et al., 2015), or domain shift (Izmailov et al., 2018; Cha et al., 2021). Several strategies have been proposed to bolster robustness against these corruptions, including augmentation-based (Hendrycks et al., 2021b;a; 2022), model-based (Kim et al., 2021; Mao et al., 2021; He et al., 2021), and adaptation-based (Wang et al., 2021; Rusak et al., 2022) methods. Among them, augmentation-based strategies are desirable for their wide applicability across different contexts. Nonetheless, existing augmentation methods, while effective against common corruptions, are limited by fixed image operations and often falter against various distributional shifts including adversarial attacks.

To address these limitations, we introduce a strategy named Smoothing Perturbations In DEcreasing Radii, or SPIDER. SPIDER leverages label smoothing as an unconventional mechanism to address perturbation uncertainty, adjusting the confidence of the true label in relation to perturbation magnitude, thus enhancing model robustness.

The two primary contributions of our paper are:

- We propose a novel interpretation of label smoothing as a method for incorporating perturbation uncertainty. By adjusting the confidence of the true label with perturbation intensity, our approach uniquely enhances both common and adversarial robustness, addressing a limitation in existing augmentation techniques.
- We conduct a detailed theoretical and empirical analysis of how our algorithm enhances robustness by examining boundary thickness and flat minima. Boundary thickness (Yang et al., 2020) measures the model's non-overconfident probability areas, while flat minima (Izmailov et al., 2018) identifies regions with stable loss against perturbations.

The remainder of this paper is structured as follows: Section 2 discusses related works on increasing model robustness against common corruptions; Section 3 formalizes the concept of SPIDER and investigates its properties in relation to the boundary thickness and flatness both theoretically and empirically; finally, Section 4 presents our primary results, followed by ablation studies.

## 2 RELATED WORKS

**Data Augmentation-based Approaches**: These methods enhance model robustness via training data augmentation. For instance, AugMix (Hendrycks et al., 2021b) creates mew images through sequences of random image processing operations, such as cropping or shearing. PixMix (Hendrycks et al., 2022) is an extension of AugMix that additionally combines images chosen from patterned sets like fractal images. Unlike our approach, both depend on predefined image operations or image sets.

DeepAugment (Hendrycks et al., 2021a) uses a pretrained model to generate augmented images and introduces predefined modifications to the network. Adversarial Noise Training (ANT) (Rusak et al., 2020) utilizes a noise generator to create adversarial noise aimed at confusing the classifier. Both DeepAugment and ANT involve the use of an additional network, either pretrained or trained.

Mixup (Zhang et al., 2018) and CutMix (Yun et al., 2019) create new training samples by interpolating two samples from different classes or cutting and pasting sections from different images while adjusting the labels accordingly. Although both have been known to improve model performance on vision tasks, they marginally improve model robustness due to the manifold intrusion problem.

Our algorithm, SPIDER, falls into the augmentation-based approach category. Compared to existing augmentations, SPIDER utilizes simpler augmentation techniques without requirements on a predefined set of images or other pretrained models.

**Model Architecture-based Approaches**: These strategies involve developing unique training schemes for particular models. For instance, QualNet (Kim et al., 2021) employs a two-stage invertible neural architecture that processes clean and corrupted images to enhance robustness. Vision transformer (Dosovitskiy et al., 2021) is another model architecture that has gained attention recently, with many succeeding works (Zhou et al., 2022; Mao et al., 2022a;b) modifying its components for increased robustness or designing self-supervised tasks for it. While these model-specific methods have shown remarkable performance, they often lack generalizability due to their reliance on specific backbone architectures and have limitations to be used together with other methods.

**Adaptation-based Approaches**: These methods improve model robustness using the principles of domain adaptation. Test Entropy Minimization (TENT) (Wang et al., 2021) is one such technique that tunes the parameters of the batch normalization layers in the test time. Robust Pseudo Labeling (RPL) (Rusak et al., 2022) operates in an unsupervised domain adaptation setting, and uses self-learning to train classifiers. While effective, these adaptation-based methods often require additional access to target data during either training or testing, limiting their applicability to specific circumstances.

**Label Smoothing-based Approaches**: These approaches focus on leveraging label smoothing which basically softens the one-hot labels of samples. Shafahi et al. (2019) revealed an empirical connection between the logit gap, gradient gap, and adversarial robustness, pointing out the potential benefits of integrating Gaussian noise with standard label smoothing. Similarly, Fu et al. (2020) presented empirical evidence of label smoothing's benefits in enhancing adversarial robustness. Nonetheless, these works have no theoretical understanding on how the adversarial robustness is achieved with label smoothing.

AutoLabel (Stutz et al., 2023) relates the degree of label smoothing to the strength of augmentations denoted as the distance between the augmented and the original image. Nonetheless, the distance metric is method-specific limiting its cross-method applicability. SSAT (Ren et al., 2022) incorporates label smoothing with adversarial perturbations, but have limited theoretical understandings on the behavior of perturbations. Additionally, strategies focusing on logits without considering perturbations have been suggested by Goibert & Dohmatob (2019).

When compared to our work, SPIDER, None of the above works have repurposed label smoothing as a monotonically decreasing function embedding the uncertainty of perturbations and lack theoretical understandings on how label smoothing and perturbation should be combined to improve general model robustness encompassing common corruptions.

# 3 SPIDER: SMOOTHING PERTURBATIONS IN DEcreasing RADII

## 3.1 PROPOSED ALGORITHM

Taking inspiration from ocean waves, SPIDER treats perturbed datapoints like sea waves, which lose strength as they travel from their origin (Figure 1a, left). With a given perturbation probability density function $p(\cdot)$, SPIDER mitigates perturbation uncertainty via a monotonically decreasing smoothing function $s(\cdot)$ that finetunes the true label. The influence of SPIDER on the training of neural networks is illustrated by the decision boundary of a SPIDER-trained network in Figure 1b.

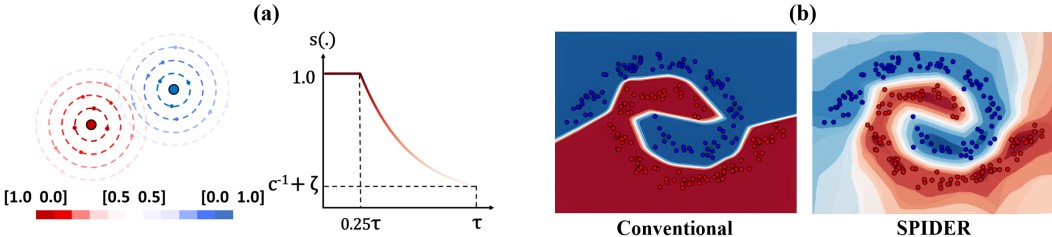

Figure 1: SPIDER methodology illustration. (a) We observe how the certainty of the true label decreases as the magnitude of perturbation increases. (b) We demonstrate the resulting smoother, contour-like decision boundary induced by the SPIDER approach compared to traditional training.

---

**Algorithm 1** SPIDER - Core Framework

---

**Input:** input datapoint $(x, y)$, isotropic perturbation pdf $p(\cdot) : \mathbb{R}^n \to \mathbb{R}$, monotonically decreasing smoothing ftn $s(\cdot) : \mathbb{R}_{\geq 0} \to \mathbb{R}_{\geq 1/c}$ (*c = number of classes*)
**Output:** augmented datapoint $(\tilde{x}, \tilde{y})$

$\tilde{x} := x + \delta, \delta \sim p$

$\tilde{y} := [\tilde{y}_1, \tilde{y}_2, \cdots, \tilde{y}_c]$, where $\tilde{y}_{i \in [c]} = \begin{cases} s(\|\delta\|) & \text{if } i \text{ is the true label} \\ (1 - s(\|\delta\|))/(c-1) & \text{otherwise} \end{cases}$

**return** $(\tilde{x}, \tilde{y})$

---

**Algorithm 2** SPIDER - Specific Instantiation

---

**Input:** input datapoint $(x, y)$, hyperparameters $\tau, \xi$
**Output:** augmented datapoint $(\tilde{x}, \tilde{y})$

$\delta = \epsilon/\|\epsilon\| \cdot r, \quad \epsilon \sim N(0, I), \quad r \sim \text{Uniform}(0, \tau)$

$s(z) := \text{clip}(e^{-\lambda(z-\epsilon)}, 0, 1), \quad \lambda := (3/4\tau)^{-1} \ln c/(1 + \xi c)$

$\tilde{x} := x + \delta, \delta \sim p$

$\tilde{y} := [\tilde{y}_1, \tilde{y}_2, \cdots, \tilde{y}_c]$, where $\tilde{y}_{i \in [c]} = \begin{cases} s(\|\delta\|) & \text{if } i \text{ is the true label} \\ (1 - s(\|\delta\|))/(c-1) & \text{otherwise} \end{cases}$

**return** $(\tilde{x}, \tilde{y})$

---

We will initially delve into the details of the chosen perturbation and smoothing functions in SPIDER (see Algorithm 2), and then elaborate on the heuristic reasoning behind these choices. Our perturbation function is akin to random sampling from an $L_2$ sphere ($\epsilon/|\epsilon|$) with a random radius $r$. The smoothing function $s(\cdot)$, conversely, is an exponentially decreasing function that sustains the confidence level of the perturbed datapoint until a certain distance ($\tau/4$) is reached (Figure 1a, right).

We opted for specific perturbation and smoothing functions based on empirical and theoretical heuristics. Our chosen perturbation function outperformed isotropic Gaussian and random $L_2$ ball sampling, which can be partly explained by the high-dimensional input space (Aggarwal et al., 2001). First, in such spaces, random $L_2$ ball sampling approximates $L_2$ sphere sampling, resulting in perturbations mostly at a fixed $L_2$ distance $r$. Second, isotropic Gaussian distribution can be viewed as $\epsilon \cdot r$, where $r$ follows a skewed chi-squared distribution, potentially leading to the underrepresentation of perturbations either too distant or too proximate to the datapoint.

### 3.2 BOUNDARY THICKNESS

#### 3.2.1 UNDERSTANDING BOUNDARY THICKNESS

Boundary thickness (Yang et al., 2020) serves as a gauge for the areas of a classifier where overconfidence does not occur. It measures the distance between two data points, $x_i$ and $x_j$, with differing labels $i$ and $j$, within which the classifier's predictions remain balanced and not overconfident.

Formally, boundary thickness can be defined as:

$$\Theta(f, \alpha, \beta, x_i, x_j) := \|x_i - x_j\| \int_0^1 I\{\alpha < g_{ij}(x(t)) < \beta\}dt$$

Here, $x(t) := (1-t)x_i + tx_j$ represents a point on the line segment between $x_i$ and $x_j$. The term $g_{ij}(x(t)) := f(x(t))_i - f(x(t))_j$ signifies the difference between the predicted probabilities of labels $i$ and $j$ at point $x(t)$. The parameters $\alpha$ and $\beta$ range between -1 and 1, and $I$ is an indicator function.

Classifiers with thicker boundaries are less prone to boundary tilting problem (Tanay & Griffin, 2016), a phenomenon in which the decision boundary leans toward one class's manifold over another's. Such imbalance makes models vulnerable to misclassifications when subjected to minor data perturbations. Theoretical and empirical evidence in the original paper validate that models with thicker boundaries improves robustness against both common and adversarial corruptions.

#### 3.2.2 SPIDER INCREASES BOUNDARY THICKNESS

Herein, we explain both theoretically and empirically how SPIDER's label smoothing function enhances a model's boundary thickness. A detailed exposition of Theorem 1 is in Appendix B.

**Theoretical Examination of SPIDER's Impact on Boundary Thickness**

**Problem Setup**: Suppose we have two distinct data points $(x_i, y_i)$ and $(x_j, y_j)$ from the training dataset. We define $\epsilon$ as a random vector in $\mathbb{R}^n$ with a probability density function (pdf) $p_\epsilon(\cdot)$ symbolizing noise, and $s(\cdot) : \mathbb{R}_{\geq 0} \to \mathbb{R}_{1/c}$ as a smoothing function mapping the $L_2$-norm of noise to a smoothed label. We consider an isotropic pdf $p_\epsilon$ centered at the origin that is formalized as $p_\epsilon(z) := p_\gamma(\|z\|)/S_n(\|z\|)$, where $p_\gamma$ defines the probability of the perturbation size and $S_n(r) := 2\pi^{\frac{n}{2}} r^{n-1} / \Gamma(n/2)$ is a function defining the surface of a $n$-dim hypersphere with radius $r$.

Consider a datapoint $(x, y)$. The perturbed version can be denoted as $(\tilde{x}, y)$, where $\tilde{x} := x + \epsilon$ without loss of generality. Furthermore, the perturbed datapoint with the smoothed label can be represented as $(\tilde{x}, \tilde{y}) := (x + \epsilon, [\tilde{y}_1, \tilde{y}_2, \cdots, \tilde{y}_c])$, with $\tilde{y}_i = s(\|\epsilon\|)$ if true label and $(1 - s(\|\epsilon\|)/(c-1)$ otherwise.

We introduce $f^*$ and $f^*_{LS}$ as optimal functions that minimize the surrogate loss along the segment $x(t) = (1-t)x_i + tx_j (t \in [0, 1])$, given the stochastic datapoints $(\tilde{x}_i, y_i), (\tilde{x}_j, y_j)$ for $f^*$ and $(\tilde{x}_i, \tilde{y}_i), (\tilde{x}_j, \tilde{y}_j)$ for $f^*_{LS}$.

**Theorem 1.** *Given any isotropic pdf $p_\epsilon(z) := p_\gamma(\|z\|)/S_n(\|z\|)$ with monotonically decreasing function $p_\gamma(\cdot)$ and a monotonically decreasing function $s(\cdot)$, the boundary thickness of $f^*_{LS}$ is always greater than or equal to $f^*$, i.e.*

$$\forall -1 \leq \alpha \leq 0 < \beta \leq 1, \ \Theta(f^*, \alpha, \beta, x_i, x_j) \leq \Theta(f^*_{LS}, \alpha, \beta, x_i, x_j)$$

*for datapoints $(x_i, y_i), (x_j, y_j)$ satisfying $y_i \neq y_j$ in the dataset.*

**Empirical Evidence for SPIDER's Enhancement of Boundary Thickness**

Our experiments comprehensively demonstrate the effectiveness of SPIDER in augmenting the boundary thickness, whether applied independently or in synergy with prior augmentation methods. We also establish the pivotal role of the label smoothing function within SPIDER in enhancing boundary thickness.

To illustrate this, we conducted experiments under two different parameter configurations: $\alpha = 0, \beta = 0.75$ and $\alpha = 0, \beta = 1$. The former configuration adheres to the default values utilized in the original boundary thickness paper (Yang et al., 2020), while the latter aligns with values known to generalize the notion of margin in Support Vector Machines (Hearst et al., 1998). For further experimental details, readers are referred to Appendix E.

Table 1: Boundary thickness of baseline augmentations.

| $(\alpha, \beta)$ | Baseline | CIFAR-10 | | CIFAR-100 | |
|---|---|---|---|---|---|
| | | Original | +SPIDER | Original | +SPIDER |
| (0, 0.75) | no aug. | $0.1582 \pm 0.0360$ | $0.5784 \pm 0.0747 (+0.4202)$ | $0.4763 \pm 0.0011$ | $0.6487 \pm 0.0312 (+0.1724)$ |
| | AugMix | $0.2126 \pm 0.0125$ | $0.2411 \pm 0.0429 (+0.0285)$ | $0.4490 \pm 0.0024$ | $0.5960 \pm 0.0167 (+0.1470)$ |
| | DeepAug. | $0.1497 \pm 0.0018$ | $0.3297 \pm 0.0674 (+0.1800)$ | $0.4696 \pm 0.0070$ | $0.6208 \pm 0.0456 (+0.1512)$ |
| | PixMix | $0.1293 \pm 0.0009$ | $0.1828 \pm 0.0105 (+0.0535)$ | $0.4403 \pm 0.0022$ | $0.5315 \pm 0.0268 (+0.0912)$ |
| (0, 1) | no aug. | $0.9376 \pm 0.0150$ | $1.8954 \pm 0.0408 (+0.9578)$ | $0.9041 \pm 0.0097$ | $1.4760 \pm 0.0354 (+0.5719)$ |
| | AugMix | $1.2152 \pm 0.0179$ | $1.3875 \pm 0.0994 (+0.1723)$ | $0.9698 \pm 0.0122$ | $1.2314 \pm 0.0172 (+0.2616)$ |
| | DeepAug. | $1.0459 \pm 0.0309$ | $1.7484 \pm 0.0970 (+0.7025)$ | $0.9467 \pm 0.0044$ | $1.2233 \pm 0.0068 (+0.2766)$ |
| | PixMix | $0.8217 \pm 0.0053$ | $1.1064 \pm 0.0456 (+0.2847)$ | $0.9669 \pm 0.0206$ | $1.1102 \pm 0.0165 (+0.1433)$ |

Table 2: Ablation results on the boundary thickness of SPIDER. (CIFAR-10)

| Baseline | $\alpha = 0, \beta = 0.75$ | | $\alpha = 0, \beta = 1$ | |
|---|---|---|---|---|
| | +SPIDER | +SPIDER w/o LS | +SPIDER | +SPIDER w/o LS |
| no aug. | $0.5784 \pm 0.0747$ | $0.1802 \pm 0.0016 (-0.3982)$ | $1.8954 \pm 0.0408$ | $1.3108 \pm 0.0440 (-0.5846)$ |
| AugMix | $0.2411 \pm 0.0429$ | $0.1365 \pm 0.0045 (-0.1046)$ | $1.3875 \pm 0.0994$ | $0.9738 \pm 0.2459 (-0.4137)$ |
| DeepAug | $0.3297 \pm 0.0674$ | $0.1898 \pm 0.0003 (-0.1399)$ | $1.7484 \pm 0.0970$ | $1.2421 \pm 0.0169 (-0.5063)$ |
| PixMix | $0.1828 \pm 0.0105$ | $0.1238 \pm 0.0017 (-0.0590)$ | $1.1064 \pm 0.0456$ | $0.7639 \pm 0.0092 (-0.3425)$ |

Tables 1 display the boundary thickness for baseline augmentations in comparison to SPIDER. It's evident across both CIFAR-10 and CIFAR-100 datasets that SPIDER consistently results in a significant increase in boundary thickness, regardless of the augmentation methods (no augmentation, AugMix (Hendrycks et al., 2021b), DeepAugment (Hendrycks et al., 2021a), PixMix (Hendrycks et al., 2022)) used in conjunction with it. This increase is universally observed for both the default and custom parameter configurations.

To further substantiate the role of label smoothing in SPIDER, we performed an ablation study. As shown in Table 2, when label smoothing is omitted (SPIDER w/o LS), a decrease in boundary thickness is observed in all the cases. This affirms our theoretical assertion (Theorem 1) regarding the critical role of label smoothing in enhancing boundary thickness.

## 3.3 FLATNESS

### 3.3.1 UNDERSTANDING FLATNESS

Flatness characterizes the extent of change in a model's loss across proximate points in the parameter or input space. Flatness-aware minimization techniques have been widely adopted in the field of domain generalization (Foret et al., 2021; Izmailov et al., 2018; Cha et al., 2021), where flatter minima contribute to better performance. This can be formally presented as a min-max problem:

$$\min_{\theta} \max_{\|\Delta\| \leq \gamma} \mathbb{E}_{(x,y) \sim D}[L(f(x; \theta + \Delta), y)]$$

Here, $\Delta$ denotes the neighborhood of point $\theta$ within a radius $\gamma$, while $D$ stands for a given dataset and $L$ for a surrogate loss function.

Flat minima is proven to improve model's ability to perform well in the face of distributional shifts in the data both theoretically and empirically (Cha et al., 2021). The distributional shifts not only include domain shifts but also data corruptions such as common corruptions and adversarial corruptions as well (Stutz et al., 2021).

Assuming a relationship between input perturbation and parameter perturbation, we can infer that minimizing surrogate loss with perturbed datapoints is akin to finding flat minima in the parameter space. Therefore, as a perturbation-based algorithm, SPIDER is also expected to favor flat minima. In subsequent sections, we will first demonstrate theoretically that perturbations in the input and parameter spaces are interconnected, and then empirically show that SPIDER indeed promotes flatter minima in the parameter space.

### 3.3.2 SPIDER ENCOURAGES FLAT MINIMA

Developing a closed-form solution to connect perturbations in input and parameter spaces for a universal classifier $f : \mathbb{R}^n \to \mathbb{R}^m$ is infeasible. As a workaround, we address the case where $f$ is a single-layer network with softmax activation, denoted as $f(x; W)$, with $W \in \mathbb{R}^{m \times n}$. We first theoretically prove the relationship between perturbations in input and parameter spaces, then empirically demonstrate that SPIDER indeed facilitates flatter minima in parameter space.

**Translating Perturbations Between Input and Parameter Spaces**

**Problem Setup:** Given a linear model $f : x \mapsto \sigma(Wx + b)$, our aim is to understand the linkage between an input perturbation($\delta$) and a parameter perturbation ($\Delta$) that satisfy $\sigma(W(x + \delta) + b) = \sigma((W + \Delta)x + b)$. Given an input perturbation bounded by the $L_2$-norm, i.e., $|\delta| \leq \gamma$, what is the potential perturbation region $R_\Delta$ for $\Delta$ that ensures, for any $|\delta| \leq \gamma$, there is a $\Delta \in R_\Delta$ fulfilling the equality, or vice versa? Similarly, what is the perturbation region $R_\delta$ for $\delta$, provided $|\Delta| \leq \gamma$?

**Definition 1.** *(Definition of $R_\delta$) Given $W \in \mathbb{R}^{m \times n}$, $x \in \mathbb{R}^n$, and a parameter perturbation region $\{\Delta \in \mathbb{R}^{m \times n} \mid \|\Delta\| \leq \gamma\}$, $R_\delta \in \mathbb{R}^n$ is a region satisfying the following constraint:*

$$\forall \|\Delta\| \leq \gamma, \exists \delta \in R_\delta \ \ s.t. \ W\delta = \Delta x \text{ and } \forall \delta \in R_\delta, \exists \|\Delta\| \leq \gamma \ \ s.t. \ W\delta = \Delta x$$

**Definition 2.** *(Definition of $R_\Delta$) Given $W \in \mathbb{R}^{m \times n}$, $D = \{x_1, \cdots, x_N\}(x_i \in \mathbb{R}^n/\{0\}$ for $i \in [N])$, and input perturbation region $\{\delta \in \mathbb{R}^n \mid \|\delta\| \leq \gamma\}$, $R_\Delta \in \mathbb{R}^{m \times n}$ is a region that satisfies the following constraint:*

$$\forall x \in D, \forall \|\delta\| \leq \gamma, \exists \Delta \in R_\Delta \ s.t. \ W\delta = \Delta x \text{ and } \forall x \in D, \forall \Delta \in R_\Delta, \exists \|\delta\| \leq \gamma \ s.t. \ W\delta = \Delta x$$

We have defined the regions of interest. Now we present theorems on the translation of perturbations from input space to parameter space (Theorem 2) and the inverse (Theorems 3 and 4). Informally, Theorem 2 suggests that parameter space perturbations bounded by $L_2$ norm can be translated to input space perturbations within a rotated ellipsoid. However, translating the perturbation from the input space to parameter space in a closed form expression is infeasible. Instead, we offer the subset and the superset of the converted perturbation region in hyperparameter space in Theorems 3 and 4. The square matrix $X_\lambda$ of dimensions $(m \times n)^2$ is defined by the input $x$ and weight $W$. The formal definition of $X_\lambda$ is given in Appendix D.

**Theorem 2.** *Given $W \in \mathbb{R}^{m \times n}$, $x \in \mathbb{R}^n$, and parameter perturbation region $\{\Delta \in \mathbb{R}^{m \times n} \mid \|\Delta\| \leq \gamma\}$, a m-dim rotated ellipsoid satisfies the definition of $R_\delta$.*

**Theorem 3.** *Given $W \in \mathbb{R}^{m \times n}$, $D = \{x_1, \cdots, x_N\}(x_i \in \mathbb{R}^n/\{0\}$ for $i \in [N])$, and input perturbation region $\{\delta \in \mathbb{R}^n \mid \|\delta\| \leq \gamma\}$, let $x_{max} := \arg\max_{x_i} \|x_i\|$ and $\lambda_{min} := \min\{\lambda_1, \cdots, \lambda_m\}$. Then, $\{\Delta \in \mathbb{R}^{m \times n} \mid \|\Delta\| \leq (\|x_{max}\|^2/\lambda_{min}^2)^{-1}\}$ is the subset of $R_\Delta$.*

**Theorem 4.** *Given $W \in \mathbb{R}^{m \times n}$, $D = \{x_1, \cdots, x_N\}(x_i \in \mathbb{R}^n/\{0\}$ for $i \in [N])$, and input perturbation region $\{\delta \in \mathbb{R}^n \mid \|\delta\| \leq \gamma\}$, let $R_i := \{d \in \mathbb{R}^{m \times n} \mid d^\top X_\lambda^{(i)} d \leq 1\}$ and $\Gamma := \{R_i \mid i \in [N]\}$. Then, $\{\arg\min_{R_1, \cdots, R_n \in \Gamma} \max_{\rho \in \cup_{i \in [n]} R_i} \|\rho\|^2\}$ is the superset of $R_\Delta$.*

**Empirical Evaluation on the Flatness of SPIDER**

We utilized the Monte-Carlo method by Cha et al. (2021) to assess flatness, applying random perturbations to a fully trained parameter vector $\theta$ and computing the average loss for those perturbations. To elaborate, we measure $\mathbb{E}_{(x,y) \sim D}[L(f(x; \theta + \Delta), y)]$, where $\|\Delta\| = \gamma$ with $\gamma$ as an increasing radius. We conducted a robust Monte-Carlo simulation, including 150 individual samples for each method to ensure the reliability of our findings.

Figure 2 visually summarizes our findings. We started from a radius of 30 for clean data, DeepAugment, and AugMix, and 50 for PixMix, increasing in 2.5 steps. Loss values for baseline methods and those combined with SPIDER were plotted on a logarithmic scale. In eight scenarios tested, SPIDER consistently found flatter minima in seven when combined with baselines. The only outlier was AugMix with SPIDER, as AugMix had already achieved flat minima compared to other CIFAR-10 augmentations. This supports our hypothesis that SPIDER, a perturbation-based algorithm, favors flatter minima. Further details of our experimental setup and results can be found in Appendix E.

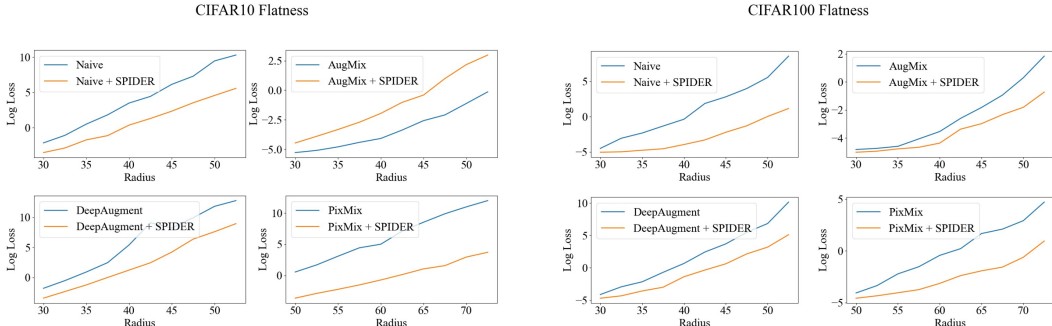

Figure 2: Flatness measured with respect to increasing radius.

# 4 EXPERIMENTS

## 4.1 EVALUATION METHODOLOGY

**General Approach:** We evaluate model robustness on CIFAR-10, CIFAR-100, Tiny-ImageNet, and ImageNet (Deng et al., 2009). Metrics for common corruption error, adversarial error, and clean accuracy are averaged over three runs for all datasets except ImageNet, which is assessed based on a single run due to computational constraints. Hyperparameters for SPIDER ($\tau$, $\xi$) are fine-tuned over 30 trials using Optuna (Akiba et al., 2019) with a TPE sampler (Bergstra et al., 2011) except for Tiny-ImageNet and ImageNet experiments fine-tuned over 15 trials. We report the models exhibiting the highest accuracy on the augmented validation set, then evaluate these models' robustness on common corruption/adversarial corruption benchmarks.

**Common Corruption and Adversarial Robustness Metrics:** We use CIFAR-10/100-C, Tiny-ImageNet-C, and ImageNet-C (Hendrycks & Dietterich, 2019) benchmarks to evaluate common corruption robustness, measuring the mean corruption error (mCE) for each baseline and dataset. For adversarial robustness, we employ untargeted $L_2$ and $L_\infty$ norm-based PGD attacks (Madry et al., 2018) with specific parameters for each dataset. For additional details, refer to Appendix E.

## 4.2 EMPIRICAL EVALUATION ON THE ROBUSTNESS OF SPIDER

Table 3: Evaluation of robustness on CIFAR-10/100 and Tiny-ImageNet Benchmarks.

| Benchmark | Method | mCE ↓ | | $L_2$ (PGD) ↓ | | $L_\infty$ (PGD) ↓ | |
|---|---|---|---|---|---|---|---|
| | | original | +SPIDER | original | +SPIDER | original | +SPIDER |
| CIFAR-10 | no aug. | $25.57 _{\pm 0.45}$ | $14.37 _{\pm 0.31}$ | $68.57 _{\pm 2.47}$ | $23.76 _{\pm 0.84}$ | $99.52 _{\pm 0.30}$ | $56.30 _{\pm 3.23}$ |
| | AugMix | $11.83 _{\pm 0.05}$ | $9.17 _{\pm 0.11}$ | $36.91 _{\pm 0.09}$ | $30.98 _{\pm 2.65}$ | $93.15 _{\pm 2.14}$ | $72.14 _{\pm 9.77}$ |
| | DeepAug. | $13.21 _{\pm 0.11}$ | $11.18 _{\pm 0.16}$ | $54.86 _{\pm 2.32}$ | $27.92 _{\pm 0.42}$ | $95.86 _{\pm 1.10}$ | $71.32 _{\pm 2.16}$ |
| | PixMix | $11.86 _{\pm 0.10}$ | $8.36 _{\pm 0.02}$ | $65.37 _{\pm 0.10}$ | $36.72 _{\pm 3.35}$ | $99.69 _{\pm 0.06}$ | $84.44 _{\pm 5.78}$ |
| CIFAR-100 | no aug. | $52.21 _{\pm 0.47}$ | $38.62 _{\pm 0.36}$ | $93.25 _{\pm 0.15}$ | $62.62 _{\pm 0.44}$ | $99.58 _{\pm 0.09}$ | $92.76 _{\pm 0.60}$ |
| | AugMix | $37.80 _{\pm 0.24}$ | $33.56 _{\pm 0.06}$ | $91.76 _{\pm 0.20}$ | $71.00 _{\pm 0.98}$ | $99.75 _{\pm 0.02}$ | $96.16 _{\pm 0.38}$ |
| | DeepAug. | $39.41 _{\pm 0.23}$ | $34.64 _{\pm 0.12}$ | $88.83 _{\pm 0.39}$ | $68.93 _{\pm 0.20}$ | $99.41 _{\pm 0.12}$ | $95.40 _{\pm 0.25}$ |
| | PixMix | $38.15 _{\pm 0.49}$ | $30.98 _{\pm 0.22}$ | $92.86 _{\pm 0.18}$ | $75.46 _{\pm 1.29}$ | $99.78 _{\pm 0.15}$ | $97.12 _{\pm 0.22}$ |
| Tiny-ImageNet | no aug. | $74.58 _{\pm 0.03}$ | $69.57 _{\pm 0.17}$ | $57.11 _{\pm 0.27}$ | $49.29 _{\pm 0.43}$ | $99.24 _{\pm 0.07}$ | $97.27 _{\pm 0.09}$ |
| | AugMix | $65.02 _{\pm 0.06}$ | $61.99 _{\pm 0.07}$ | $60.96 _{\pm 0.50}$ | $56.40 _{\pm 0.36}$ | $99.55 _{\pm 0.03}$ | $99.24 _{\pm 0.09}$ |
| | DeepAug. | $62.51 _{\pm 0.62}$ | $59.94 _{\pm 0.39}$ | $53.35 _{\pm 0.22}$ | $48.22 _{\pm 0.19}$ | $99.49 _{\pm 0.07}$ | $99.26 _{\pm 0.02}$ |
| | PixMix | $61.94 _{\pm 0.14}$ | $60.62 _{\pm 0.19}$ | $63.00 _{\pm 0.30}$ | $61.39 _{\pm 3.15}$ | $99.64 _{\pm 0.02}$ | $99.03 _{\pm 0.13}$ |

In our exploration of SPIDER's performance against common corruptions and adversarial attacks, we compared it with established augmentation strategies such as the absence of any augmentation (no aug.), AugMix (Hendrycks et al., 2021b), DeepAugment (Hendrycks et al., 2021a), and PixMix (Hendrycks et al., 2022). We found SPIDER consistently diminished errors with minimal to zero compromises on clean accuracy.

Table 4: Clean accuracy on CIFAR-10/100 and Tiny-ImageNet benchmarks.

| Benchmark | No Augmentation | | AugMix | | DeepAugment | | PixMix | |
|---|---|---|---|---|---|---|---|---|
| | original | +SPIDER | original | +SPIDER | original | +SPIDER | original | +SPIDER |
| CIFAR10 | $95.37_{\pm 0.06}$ | $94.56_{\pm 0.14}$ | $95.78_{\pm 0.07}$ | $95.33_{\pm 0.08}$ | $94.90_{\pm 0.17}$ | $92.28_{\pm 0.39}$ | $95.52_{\pm 0.19}$ | $95.87_{\pm 0.09}$ |
| CIFAR100 | $76.76_{\pm 0.09}$ | $74.08_{\pm 0.16}$ | $76.85_{\pm 0.19}$ | $75.49_{\pm 0.25}$ | $74.66_{\pm 0.04}$ | $72.63_{\pm 0.37}$ | $76.81_{\pm 0.30}$ | $77.00_{\pm 0.25}$ |
| Tiny-IN | $64.74_{\pm 0.24}$ | $67.40_{\pm 0.20}$ | $65.08_{\pm 0.15}$ | $68.87_{\pm 0.28}$ | $63.98_{\pm 2.47}$ | $66.69_{\pm 0.05}$ | $66.97_{\pm 0.04}$ | $68.95_{\pm 0.23}$ |

Table 5: Evaluation of robustness and clean accuracy on ImageNet benchmark.

| Method | mCE $\downarrow$ | | $L_2$ (PGD) $\downarrow$ | | $L_\infty$ (PGD) $\downarrow$ | | clean acc. $\uparrow$ | |
|---|---|---|---|---|---|---|---|---|
| | original | +SPIDER | original | +SPIDER | original | +SPIDER | original | +SPIDER |
| no aug. | 88.17 | 81.61 | 76.96 | 75.92 | 98.29 | 97.19 | 70.12 | 71.30 |
| AugMix | 84.78 | 83.25 | 78.52 | 70.90 | 98.28 | 96.33 | 69.84 | 76.91 |
| DeepAug. | 87.54 | 85.49 | 76.93 | 67.49 | 98.26 | 96.23 | 75.79 | 76.19 |
| PixMix | 78.17 | 76.72 | 77.14 | 74.64 | 98.03 | 97.93 | 74.56 | 75.17 |

\* The $\downarrow$ symbol denotes that lower values are better, and $\uparrow$ higher the better.

To illustrate, consider the CIFAR-10 dataset (Table 3). Without any augmentation, the mean corruption error (mCE) was found to be 25.57. However, the introduction of SPIDER reduced it significantly to 14.37. Similarly, the $L_2$ and $L_\infty$ adversarial errors, which were initially 68.57 and 99.52 respectively, dropped to 23.76 and 56.30 with the addition of SPIDER. When integrated with other augmentation strategies, SPIDER's effectiveness becomes more evident. For instance, pairing SPIDER with AugMix led to a notable reduction in mCE from 11.83 to 9.17.

In terms of clean accuracy, SPIDER's integration with various augmentations showed minimal to no compromises, and even improvements in datasets like Tiny-ImageNet and ImageNet. Particularly, while in CIFAR benchmarks SPIDER exhibited a minor decline, the overall trend underscores SPIDER's capacity to boost robustness without sacrificing and sometimes enhancing clean accuracy.

Interestingly, while previous augmentation strategies for common corruption robustness showed limited efficacy in adversarial robustness, SPIDER presents a contrast. As demonstrated in preceding sections, SPIDER not only enhances boundary thickness but also fosters flatter minima, leading to improved robustness against general distributional shifts. Our experiments underscore its strength: it bolstered both common and adversarial robustness across all evaluation scenarios.

In summary, our study showcases the efficacy of SPIDER in providing robustness against both common and adversarial corruptions across multiple benchmarks and augmentation strategies. Its consistent performance, coupled with its minimal impact on clean accuracy, makes it a compelling addition to the toolbox of techniques for enhancing model robustness.

## 4.3 ABLATION STUDY

Within this section, we aim to discern the individual contributions of SPIDER's components to its robustness on CIFAR-100 benchmarks. At its core, SPIDER operates on a dual-axis: the perturbation function and the label smoothing function. These components are engineered to collaboratively bolster model robustness, but how do they fare independently?

We first tested the scenario where we retained the perturbations but disabled the label smoothing (denoted as 'No LS'). It provides insights into how fundamental the label smoothing mechanism is to SPIDER's overall performance. In a reciprocal experiment, we implemented the standard label smoothing without SPIDER's perturbations (denoted as 'STD LS'). For clarity, standard label smoothing (Szegedy et al., 2016) operates by tempering the probability of the correct class and uniformly allocating the residual probabilities across other classes. By isolating these functions, we sought to understand how much each contributes to the robustness in isolation and whether there's any potential redundancy in their combined use. Detailed experimental procedures can be found in Appendix E.

Table 6: Ablation results of model robustness on CIFAR-100 benchmarks.

| Metric | Method | Baseline | SPIDER | No LS | STD LS | Mixup |
|---|---|---|---|---|---|---|
| mCE ↓ | no aug. | $52.21 \pm 0.47$ | $38.62 \pm 0.36$ | $40.10 \pm 0.20$ | $52.48 \pm 0.30$ | $46.58 \pm 0.28$ |
| | AugMix | $37.80 \pm 0.24$ | $33.56 \pm 0.06$ | $39.68 \pm 0.15$ | $38.11 \pm 0.09$ | $35.85 \pm 0.41$ |
| | DeepAug. | $39.41 \pm 0.23$ | $34.64 \pm 0.12$ | $36.02 \pm 0.27$ | $39.72 \pm 0.05$ | $36.15 \pm 0.19$ |
| | PixMix | $38.15 \pm 0.49$ | $30.98 \pm 0.22$ | $31.59 \pm 0.29$ | $32.32 \pm 0.08$ | $32.39 \pm 0.10$ |
| $L_2 \downarrow$ | no aug. | $93.25 \pm 0.15$ | $62.62 \pm 0.44$ | $59.00 \pm 0.09$ | $68.06 \pm 0.31$ | $91.32 \pm 0.60$ |
| | AugMix | $91.76 \pm 0.20$ | $71.00 \pm 0.98$ | $72.57 \pm 0.52$ | $91.77 \pm 0.24$ | $86.83 \pm 0.84$ |
| | DeepAug. | $88.83 \pm 0.39$ | $68.93 \pm 0.20$ | $74.04 \pm 0.38$ | $69.16 \pm 0.46$ | $79.97 \pm 0.74$ |
| | PixMix | $92.86 \pm 0.75$ | $75.46 \pm 1.29$ | $75.78 \pm 0.50$ | $69.24 \pm 0.85$ | $80.42 \pm 0.16$ |
| $L_\infty \downarrow$ | no aug. | $99.58 \pm 0.09$ | $92.76 \pm 0.60$ | $96.86 \pm 0.20$ | $82.95 \pm 1.72$ | $99.84 \pm 0.05$ |
| | AugMix | $99.75 \pm 0.02$ | $96.16 \pm 0.38$ | $99.32 \pm 0.10$ | $99.72 \pm 0.05$ | $99.65 \pm 0.09$ |
| | DeepAug. | $99.41 \pm 0.12$ | $95.40 \pm 0.25$ | $99.14 \pm 0.04$ | $84.34 \pm 0.60$ | $98.31 \pm 0.36$ |
| | PixMix | $99.78 \pm 0.15$ | $97.12 \pm 0.22$ | $99.18 \pm 0.03$ | $87.48 \pm 0.72$ | $98.72 \pm 0.15$ |

Analyzing the results shown in Table 6, it became evident that removing the label smoothing function, i.e., the 'No LS' case, has a significant damage on robustness in most of the cases, suggesting its pivotal role in SPIDER's mechanism. This observation correlates well with Theorem 1, emphasizing the importance of boundary thickness and its influence on model robustness. The sole use of standard label smoothing, i.e., the 'STD LS' case, displays limited effect against common corruptions, but works well particularly against certain adversarial challenges, especially the $L_\infty$ attacks. However, when both elements are integrated as in SPIDER, the results show a more comprehensive robustness across different distributional shifts.

Our study also evaluated Mixup's (Zhang et al., 2018) role in robustness enhancement. Echoing findings from AugMix (Hendrycks et al., 2021b), techniques like CutMix (Yun et al., 2019) or Mixup often did not enhance robustness. Despite Mixup's conceptual similarity with some of our investigated methods, its contribution to robustness appeared marginal at best, and at times even counterproductive. This might resonate with AugMix's observations, where they attributed the potential subpar performance to the possible manifold intrusions.

## 5 LIMITATIONS AND FUTURE DIRECTIONS

**Perturbation Sensitivity**: Oversized perturbations risk causing datapoints to intrude into submanifolds of different labels, thereby impairing clean accuracy. On the other hand, very slight perturbations fail to enhance robustness.

**Hyperparameter Introduction**: With SPIDER comes additional hyperparameters for shaping the perturbation and smoothing functions, potentially increasing the training cost due to hyperparameter search. We have attempted to alleviate this by adopting an automated hyperparameter searching algorithm coupled with early stopping, or 'pruning'.

**Function Shape Understanding**: The relationship between model robustness and the shapes of perturbation and smoothing functions remains underexplored. While our findings confirm that isotropic perturbation functions coupled with monotonically decreasing smoothing contribute to enhanced robustness, the optimal function shapes remain undetermined. This presents avenues for future research to discern efficacious configurations to optimize SPIDER's robustness capabilities.

## 6 CONCLUSION

We present Smoothing Perturbations In DEcreasing Radii (SPIDER), a novel algorithm drawing inspiration from ocean wave energy dissipation, that refines label confidence based on perturbation magnitude. This adjustment bolsters robustness against various corruptions. We have thoroughly investigated how each component of SPIDER perturbation and smoothing function encourages larger boundary thickness and flatter minima, resulting in improvements on general model robustness.

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

# A    APPENDIX

The appendix contains additional material that could not be incorporated into the main paper due to page constraints. This includes detailed proofs of the theorems, supporting visual representations, and supplementary experimental data. The formal proofs, accompanied by illustrative diagrams that elucidate the basic concept behind each theorem, are discussed in sections B, C, and D. Section E delivers in-depth experimental methodologies that underpin the primary paper and some minor complementary experiments. These supplementary experiments comprise sensitivity analysis of the hyperparameters and a showcase of how Optuna's early stopping (pruning) algorithm has contributed to minimizing training time.

# B    DETAILED PROOFS FOR THEOREM 1 AND ACCOMPANYING LEMMAS

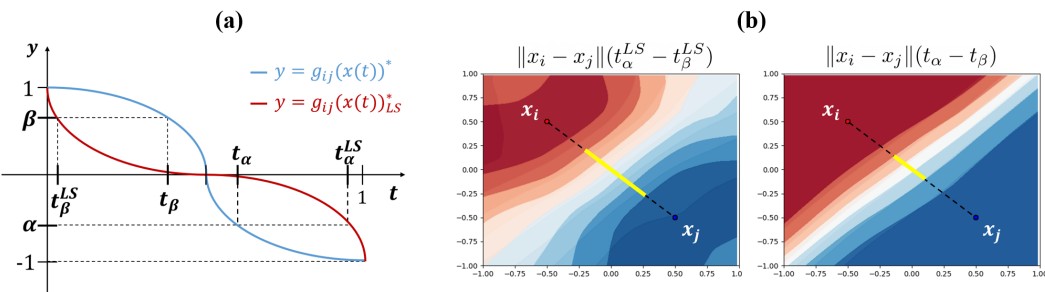

Figure 3: Depictions related to Theorem 1. **(a)** In simplified terms, the value of $g_{ij}(x(t))^*_{LS}$ for $t \in [0, 1/2]$ is lower than or equal to $g_{ij}(x(t))^*$, and the inverse holds true for $t \in [1/2, 1]$. This particular pattern gives rise to $t^{LS}_\alpha - t^{LS}_\beta \geq t_\alpha - t_\beta$, resulting in increased boundary thickness. **(b)** A simplified representation of how SPIDER's monotonically decreasing smoothing function assures increased boundary thickness. The yellow lines indicate the boundary thickness for SPIDER (on the left) and SPIDER without the smoothing function (on the right), respectively.

## B.1    PROOF ON THEOREM 1

**Theorem 1.** *Given any isotropic pdf $p_\epsilon(z) := p_\gamma(\|z\|)/S_n(\|z\|)$ with monotonically decreasing function $p_\gamma(\cdot)$ and a monotonically decreasing function $s(\cdot)$, the boundary thickness of $f^*_{LS}$ is always greater than or equal to $f^*$, i.e.*

$$\forall -1 \leq \alpha \leq 0 < \beta \leq 1, \ \Theta(f^*, \alpha, \beta, x_i, x_j) \leq \Theta(f^*_{LS}, \alpha, \beta, x_i, x_j)$$

*for datapoints $(x_i, y_i), (x_j, y_j)$ satisfying $y_i \neq y_j$ in the dataset.*

*Proof.* Let $\mathcal{E}_{lin}$ denote the event that perturbed images $x_i + \epsilon_i$ and $x_j + \epsilon_j$ lies on segment $x(t)$. Given such event $\mathcal{E}_{lin}$, we can formulate the perturbed images as $x'_i := x_i + (x_j - x_i)\eta_i$ and $x'_j = x_j + (x_i - x_j)\eta_j$, where $\eta_i$ and $\eta_j$ are univariate random variables in $[0, 1]$. Since $\epsilon_i, \epsilon_j \sim p_\epsilon(\cdot)$ where $p_\epsilon(\cdot)$ is an isotropic decreasing function centered at origin, $\eta_i, \eta_j \sim p_\eta(\cdot)$ where $p_\eta(\cdot)$ is a 1D monotonically decreasing pdf defined in the domain $[0, 1]$. In addition, we assume that $\forall t \in [0, 1], \ p_\eta(t) > 0$, i.e. the probabilities of perturbed images are nonzero at the segment $x(t)(t \in [0, 1])$ to make the calculation of $f^*$ and $f^*_{LS}$ possible. One justification for the assumption is that for small $h > 0$, $(1 - h)p_\eta(t) + h \cdot \text{Uniform}(0, 1) \approx p_\eta(t)$ in practice.

Now, we are ready to calculate the expected labels on the line segment. Since the primary interest here is boundary thickness which is parameterized by $\|x_i - x_j\|$ and $g_{ij}(x(t))$, we take a closer look at the $i$th and $j$th element of the one-hot encoded labels.

For any point $x(t)$ and its corresponding label $y = [y_0 \cdots y_i \cdots y_j \cdots y_c]$ w.l.o.g.,

$\mathbb{E}[y \mid x(t), \mathcal{E}_{lin}]$

$\propto \mathbb{E}[y \mid x_i + (x_j - x_i)\eta_i = x(t)] \cdot p_\eta(t) + \mathbb{E}[y \mid x_j + (x_i - x_j)\eta_j = x(t)] \cdot p_\eta(1 - t)$

$$= [0 \cdots 1 \cdots 0 \cdots 0] \cdot p_\eta(t) + [0 \cdots 0 \cdots 1 \cdots 0] p_\eta(1-t)$$

$$\mathbb{E}[y \mid x(t), \mathcal{E}_{lin}] = \left[ 0 \cdots \frac{p_\eta(t)}{p_\eta(t) + p_\eta(1-t)} \cdots \frac{p_\eta(1-t)}{p_\eta(t) + \_\eta(1-t)} \cdots 0 \right]$$

, i.e. $y_i = \dfrac{p_\eta(t)}{p_\eta(t) + p_\eta(1-t)}$, $y_j = \dfrac{p_\eta(1-t)}{p_\eta(t) + p_\eta(1-t)}$, and 0 elsewhere.

Let $f^* : \mathbb{R}^n \to \Delta_c$ be any function satisfying $f^*(x(t))_i = \dfrac{p_\eta(t)}{p_\eta(t) + p_\eta(1-t)}$ and $f^*(x(t))_j = \dfrac{p_\eta(1-t)}{p_\eta(t) + p_\eta(1-t)}$ for all $t \in [0,1]$, where $\Delta_c$ is a probability simplex in $\mathbb{R}^c$. $f^*$ is an optimal function that minimizes surrogate loss along the line segment $x(t), t \in [0,1]$.

Analogously, we can formulate perturbed images with smoothed labels given $\mathcal{E}_{lin}$ as $(x_i + (x_j - x_i)\eta_i, \tilde{y}_i)$ and $(x_j + (x_i - x_j)\eta_j, \tilde{y}_j)$. The expected label $\tilde{y}$ of a point $x(t)$ on the line segment is

$$\mathbb{E}[\tilde{y} \mid x(t), \mathcal{E}_{lin}]$$

$$\propto \mathbb{E}[\tilde{y} \mid x_i + (x_j - x_i)\eta_i = x(t)] \cdot p_\eta(t) + \mathbb{E}[y \mid x_j + (x_i - x_j)\eta_j = x(t)] \cdot p_\eta(1-t)$$

$$= [r_i \cdots s(\|(x_j - x_i)\eta_i\|) \cdots r_i \cdots r_i] \cdot p_\eta(t) + [r_j \cdots r_j \cdots s(\|(x_j - x_i)\eta_j\|) \cdots r_j] p_\eta(1-t)$$

$$\left( r_i := \frac{1 - s(\|(x_j - x_i)\eta_i\|)}{c - 1}, r_j := \frac{1 - s(\|(x_i - x_j)\eta_j\|)}{c - 1} \right)$$

$$= [r_i \cdots s(\|(x_j - x_i)t\|) \cdots r_i \cdots r_i] \cdot p_\eta(t) + [r_j \cdots r_j \cdots s(\|(x_j - x_i)(1-t)\|) \cdots r_j] p_\eta(1-t)$$

$$(\because \eta_i = t \text{ and } \eta_j = 1 - t \text{ given events } x_i + (x_j - x_i)\eta_i = x(t) \text{ and } x_j + (x_i - x_j)\eta_j = x(t).)$$

For the sake of readibility, let $d_i(t) := \|(x_j - x_i)t\|$ and $d_j(t) := \|(x_i - x_j)(1-t)\|$.

$0 \le d_i(t), d_j(t) \le \|x_j - x_i\|$ and $d_i(t) + d_j(t) = \|x_j - x_i\|$. We will simply denote $d_i(t), d_j(t)$ as $d_i, d_j$, and $r_i(t), r_j(t)$ as $r_i, r_j$ wherever not necessary.

The sum of the elements of the above vector is $p_\eta(t) + p_\eta(1-t)$. Thus,

$$\mathbb{E}[\tilde{y} \mid x(t), \mathcal{E}_{lin}]$$

$$= \left[ \frac{r_i p_\eta(t) + r_j p_\eta(1-t)}{p_\eta(t) + p_\eta(1-t)} \cdots \frac{s(d_i) p_\eta(t) + r_j p_\eta(1-t)}{p_\eta(t) + p_\eta(1-t)} \right.$$
$$\left. \cdots \frac{r_i p_\eta(t) + s(d_j) p_\eta(1-t)}{p_\eta(t) + p_\eta(1-t)} \cdots \frac{r_i p_\eta(t) + r_j p_\eta(1-t)}{p_\eta(t) + p_\eta(1-t)} \right]$$

Let $f_{LS}^* : \mathbb{R}^n \to \Delta_c$ be any function satisfying

$$\forall t \in [0,1], \quad f_{LS}^*(x(t))_{k \in [c]} = \begin{cases} \dfrac{s(d_i) p_\eta(t) + r_j p_\eta(1-t)}{p_\eta(t) + p_\eta(1-t)} & \text{if } k = i, \\[3mm] \dfrac{r_i p_\eta(t) + s(d_j) p_\eta(1-t)}{p_\eta(t) + p_\eta(1-t)} & \text{if } k = j, \\[3mm] \dfrac{r_i p_\eta(t) + r_j p_\eta(1-t)}{p_\eta(t) + p_\eta(1-t)} & \text{otherwise.} \end{cases}$$

$f_{LS}^*$ is an optimal function that minimizes surrogate loss along the segment $x(t), t \in [0,1]$.

We can now calculate the boundary thickness of $f^*$ and $f_{LS}^*$. To clearly distinguish $g_{ij}(x(t))$ for $f^*$ and $f_{LS}^*$, we use notations $g_{ij}(x(t))^*$ and $g_{ij}(x(t))_{LS}^*$.

$$\Theta(f^*, \alpha, \beta, x_i, x_j) := \|x_i - x_j\| \int_0^1 I\{\alpha < g_{ij}(x(t))^* < \beta\} dt$$

$$\Theta(f_{LS}^*, \alpha, \beta, x_i, x_j) := \|x_i - x_j\| \int_0^1 I\{\alpha < g_{ij}(x(t))_{LS}^* < \beta\} dt$$

$$g_{ij}(x(t))^* = \frac{p_\eta(t)}{p_\eta(t) + p_\eta(1-t)} - \frac{p_\eta(1-t)}{p_\eta(t) + p_\eta(1-t)} = \frac{p_\eta(t) - p_\eta(1-t)}{p_\eta(t) + p_\eta(1-t)}$$

$$g_{ij}(x(t))^*_{LS} = \frac{s(d_i)p_\eta(t) + r_j p_\eta(1-t)}{p_\eta(t) + p_\eta(1-t)} - \frac{r_i p_\eta(t) + s(d_j)p_\eta(1-t)}{p_\eta(t) + p_\eta(1-t)}$$

$$= \frac{(s(d_i) - r_i)p_\eta(t) - (s(d_j) - r_j)p_\eta(1-t)}{p_\eta(t) + p_\eta(1-t)}$$

$$s(d_i) - r_i = s(d_i) - \frac{1 - s(d_i)}{c - 1} = \frac{c \cdot s(d_i) - 1}{c - 1}$$

Since $1/c \leq s(d_i) \leq 1$, $0 \leq \dfrac{c \cdot s(d_i) - 1}{c - 1} \leq 1$, i.e. $0 \leq s(d_i) - r_i \leq 1$.

For the sake of simplicity, let $\gamma_i(d_i) := s(d_i) - r_i$. $r_i(d_i)$ is a monotonically decreasing function with respect to $d_i$ ($\because s(\cdot)$ and $c \cdot s(\cdot) - 1$ are monotonically decreasing functions.) Trivially, $0 \leq \gamma_j(d_j) \leq 1$ and $\gamma_j(\cdot)$ is a monotonically decreasing function.

Recap that $g_{ij}(x(t))^* = \dfrac{p_\eta(t) - p_\eta(1-t)}{p_\eta(t) + p_\eta(1-t)}$. Let us rewrite $g_{ij}(x(t))^*_{LS}$ as:

$$g_{ij}(x(t))^*_{LS} = \frac{\gamma_i(d_i)p_\eta(t) - \gamma_j(d_j)p_\eta(1-t)}{p_\eta(t) + p_\eta(1-t)}$$

One important thing to note here is that both $g_{ij}(x(t))^*$ and $g_{ij}(x(t))^*_{LS}$ are monotonically decreasing functions with respect to $t$ that are symmetric to the point $(1/2, 0)$.

**Lemma 1.** *$g_{ij}(x(t))^*$ and $g_{ij}(x(t))^*_{LS}$ are monotonically decreasing functions with respect to $t \in [0, 1]$ that are symmetric to the point $(1/2, 0)$.*

Another thing to note is that if we show $0 \leq g_{ij}(x(t))^*_{LS} \leq g_{ij}(x(t))^*$ for $t \in [0, 1/2]$ and $0 \geq g_{ij}(x(t))^*_{LS} \geq g_{ij}(x(t))^*$ for $t \in [1/2, 1]$, at the same time we are showing $\forall -1 \leq \alpha \leq 0 \leq \beta \leq 1, \Theta(f^*_{LS}, \alpha, \beta, x_i, x_j) \geq \Theta(f^*, \alpha, \beta, x_i, x_j)$ considering Lemma 1 (Figure 3a.)

**Lemma 2.** *$0 \leq g_{ij}(x(t))^*_{LS} \leq g_{ij}(x(t))^*$ for $t \in [0, 1/2]$ and $0 \geq g_{ij}(x(t))^*_{LS} \geq g_{ij}(x(t))^*$ for $t \in [1/2, 1]$.*

With Lemma 1 and Lemma 2, we can now finally derive that the boundary thickness of $f^*_{LS}$ is greater than or equal to $f^*$.

Let $0 \leq t^{LS}_\beta \leq 1/2 \leq t^{LS}_\alpha \leq 1$ be the values of $t$ such that

$$(t^{LS}_\alpha, t^{LS}_\beta) = \left( \inf_t g_{ij}(x(t))^*_{LS} = \alpha, \sup_t g_{ij}(x(t))^*_{LS} = \beta \right).$$

Since $g_{ij}(x(t))^*_{LS}$ is monotonically decreasing, the boundary thickness of $f^*_{LS}$ is $\|x_i - x_j\| \int_0^1 I\{\alpha < g_{ij}(x(t))^*_{LS} < \beta\} dt = \|x_i - x_j\| \cdot (t^{LS}_\alpha - t^{LS}_\beta)$.

Similarly, let $0 \leq t_\beta \leq 1/2 \leq t_\alpha \leq 1$ be the values of $t$ such that

$$(t_\alpha, t_\beta) = \left( \inf_t g_{ij}(x(t))^* = \alpha, \sup_t g_{ij}(x(t))^*_{LS} = \beta \right).$$

The boundary thickness of $f^*$ is $\|x_i - x_j\| \cdot (t_\alpha - t_\beta)$.

We now show that $t^{LS}_\beta \leq t_\beta$ using Lemma 1 and Lemma 2, with the definition of $t^{LS}_\beta$ and $t_\beta$. For the sake of simplicity, let $h^*(t)$ and $h^*_{LS}(t)$ denote $g_{ij}(x(t))^*$ and $g_{ij}(x(t))^*_{LS}$ respectively.

$$
\begin{aligned}
& h_{LS}^*(t_\beta^{LS}) = \beta \le h^*(t_\beta^{LS}) && \text{Def. of } t_\beta^{LS}, \text{ Lemma 2} \\
& h^*(t_\beta) = \beta \ge h_{LS}^*(t_\beta) && \text{Def. of } t_\beta, \text{ Lemma 2} \\
& h_{LS}^*(t_\beta^{LS}) \ge h_{LS}^*(t_\beta) && h_{LS}^*(t_\beta^{LS}) = \beta, \beta \ge h_{LS}^*(t_\beta) \\
& t_\beta^{LS} \le t_\beta && \text{Lemma 1}
\end{aligned}
$$

Likewise, we can trivially derive $t_\alpha^{LS} \ge t_\alpha$. Then, $(t_\alpha^{LS} - t_\beta^{LS}) \ge (t_\alpha - t_\beta)$, i.e.

$$
\|x_i - x_j\|(t_\alpha^{LS} - t_\beta^{LS}) \ge \|x_i - x_j\|(t_\alpha - t_\beta). \qquad \square
$$

### B.2   PROOF ON LEMMA 1

**Lemma 1.** $g_{ij}(x(t))^*$ and $g_{ij}(x(t))^*_{LS}$ are monotonically decreasing functions with respect to $t \in [0, 1]$ that are symmetric to the point $(1/2, 0)$.

*Proof.* We divide the proof into two parts. First we prove that $g_{ij}(x(t))^*$ and $g_{ij}(x(t))^*_{LS}$ are symmetric to the point $\left(\frac{1}{2}, 0\right)$, and then prove the functions are monotonically decreasing.

A function $f$ being symmetric to a point $\left(\frac{1}{2}, 0\right)$ indicates that $f\left(\frac{1}{2} + x\right) = -f\left(\frac{1}{2} - x\right)$.

$$g_{ij}(x(t))^* = \frac{p_\eta(t) - p_\eta(1 - t)}{p_\eta(t) + p_\eta(1 - t)}$$

$$g_{ij}\left(x\left(\tfrac{1}{2} + t\right)\right)^* = \frac{p_\eta(\tfrac{1}{2} + t) - p_\eta(\tfrac{1}{2} - t)}{p_\eta(\tfrac{1}{2} + t) + p_\eta(\tfrac{1}{2} - t)} = -\frac{p_\eta(\tfrac{1}{2} - t) - p_\eta(\tfrac{1}{2} + t)}{p_\eta(\tfrac{1}{2} - t) + p_\eta(\tfrac{1}{2} + t)} = -g_{ij}\left(x\left(\tfrac{1}{2} - t\right)\right)$$

$$g_{ij}(x(t))^*_{LS} = \frac{(s(d_i(t)) - r_i(t))p_\eta(t) - (s(d_j(t)) - r_j(t))p_\eta(1 - t)}{p_\eta(t) + p_\eta(1 - t)}, \text{ where}$$

$$r_i(t) = \frac{1 - s(d_i(t))}{c - 1}, r_j(t) = \frac{1 - s(d_j(t))}{c - 1}$$

$d_i(t) = \|(x_j - x_i)t\|$, and $d_j(t) = \|(x_i - x_j)(1 - t)\| = \|(x_j - x_i)(1 - t)\|$.

We will unpack $g_{ij}(x(t))^*_{LS}$.

$$g_{ij}(x(t))^*_{LS} = \frac{1}{p_\eta(t) + p_\eta(1 - t)} \cdot \left( \frac{c \cdot s(d_i(t)) - 1}{c - 1} \cdot p_\eta(t) - \frac{c \cdot s(d_j(t)) - 1}{c - 1} \cdot p_\eta(1 - t) \right)$$

$$= \frac{1}{p_\eta(t) + p_\eta(1 - t)} \times$$

$$\left( \frac{c \cdot s(\|(x_j - x_i)t\|) - 1}{c - 1} \cdot p_\eta(t) - \frac{c \cdot s(\|(x_j - x_i)(1 - t)\|) - 1}{c - 1} \cdot p_\eta(1 - t) \right)$$

$$g_{ij}(x(\tfrac{1}{2} + t))^*_{LS}$$

$$= \frac{1}{p_\eta(\tfrac{1}{2} + t) + p_\eta(\tfrac{1}{2} - t)} \times$$

$$\left( \frac{c \cdot s(\|(x_j - x_i)(\tfrac{1}{2} + t)\|) - 1}{c - 1} \cdot p_\eta(\tfrac{1}{2} + t) - \frac{c \cdot s(\|(x_j - x_i)(\tfrac{1}{2} - t)\|) - 1}{c - 1} \cdot p_\eta(\tfrac{1}{2} - t) \right)$$

$$= -g_{ij}(x(\tfrac{1}{2} - t))^*_{LS}$$

Now, we prove $g_{ij}(x(t))^*$ and $g_{ij}(x(t))^*_{LS}$ are monotonically decreasing.

$$g_{ij}(x(t))^* = \frac{p_\eta(t) - p_\eta(1 - t)}{p_\eta(t) + p_\eta(1 - t)}$$

We show that $g_{ij}(x(t))^*$ is monotonically decreasing for all $t \in [0, 1/2]$. For the sake of simplicity, we use $p(\cdot)$ to denote $p_\eta(\cdot)$ here. $\forall\, 0 \le a \le b \le 1/2$,

$$g_{ij}(x(a))^* - g_{ij}(x(b))^*$$

$$= \frac{p(a) - p(1 - a)}{p(a) + p(1 - a)} - \frac{p(b) - p(1 - b)}{p(b) + p(1 - b)}$$

$$= \frac{2p(a)p(1 - b) - 2p(1 - a)p(b)}{(p(a) + p(1 - a))(p(b) + p(1 - b))}$$

$$\text{sign}(g_{ij}(x(a))^* - g_{ij}(x(b))^*) = \text{sign}(p(a)p(1 - b) - p(1 - a)p(b)) \ (\because p(\cdot) : [0, 1] \to \mathbb{R}_{>0})$$

$$p(a)p(1 - b) - p(1 - a)p(b)$$

$$= p(1-a)p(b) \cdot \left( \frac{p(a)p(1-b)}{p(1-a)p(b)} - 1 \right)$$

$$= p(1-a)p(b) \cdot \left( \frac{p(a)}{p(1-a)} \div \frac{p(b)}{p(1-b)} - 1 \right)$$

Since $p$ is monotonically decreasing, $p(a) \geq p(b) \geq p(1-b) \geq p(1-a) > 0$.

$$\frac{p(a)}{p(1-a)} \geq \frac{p(b)}{p(1-a)} \geq \frac{p(b)}{p(1-b)} \rightarrow \frac{p(a)}{p(1-a)} - \frac{p(b)}{p(1-b)} \geq 0.$$

$$\frac{p(a)}{p(1-a)} \div \frac{p(b)}{p(1-b)} \geq 1 \left( \because \frac{p(t)}{p(1-t)} > 0 \quad \forall\, t \in [0,1]. \right)$$

Thus, $p(a)p(1-b) - p(1-a)p(b) \geq 0$, which leads to $g_{ij}(x(a)) - g_{ij}(x(b)) \geq 0$ $(\forall 0 \leq a \leq b \leq 1/2.)$

Since $g_{ij}(x(t))^*$ is symmetric to the point $\left(\frac{1}{2}, 0\right)$, $g_{ij}(x(t))^*$ is also monotonically decreasing in $t \in \left[\frac{1}{2}, 1\right]$.

Likewise, let $\gamma_i(d_i(t)) := \dfrac{c \cdot s(d_i(t)) - 1}{c - 1}$. $\forall\, t \in [0,1]$, $0 \leq \gamma_i(d_i(t)) \leq 1$ and $\gamma_i(d_i(t))$ is a monotonically decreasing function.

$$g_{ij}(x(t))^*_{LS} = \frac{\gamma_i(d_i(t))p_\eta(t) - \gamma_i(1 - d_i(t))p_\eta(1-t)}{p_\eta(t) + p_\eta(1-t)}$$

Analogously, we can derive that $g_{ij}(x(t))^*_{LS}$ is monotonically decreasing function with trivial calculations. $\qquad\square$

## B.3 PROOF ON LEMMA 2

**Lemma 2.** $0 \le g_{ij}(x(t))^*_{LS} \le g_{ij}(x(t))^*$ *for* $t \in [0, 1/2]$ *and* $0 \ge g_{ij}(x(t))^*_{LS} \ge g_{ij}(x(t))^*$ *for* $t \in [1/2, 1]$.

*Proof.* $g_{ij}(x(t))^* = \dfrac{p_\eta(t) - p_\eta(1-t)}{p_\eta(t) + p_\eta(1-t)}$ and

$$g_{ij}(x(t))^*_{LS} = \frac{1}{p_\eta(t) + p_\eta(1-t)} \cdot \left( \frac{c \cdot s(d_i(t)) - 1}{c - 1} \cdot p_\eta(t) - \frac{c \cdot s(d - d_i(t)) - 1}{c - 1} \cdot p_\eta(1-t) \right)$$

$$g_{ij}(x(t))^* - g_{ij}(x(t))^*_{LS}$$
$$= \frac{1}{p_\eta(t) + p_\eta(1-t)} \cdot \left( \frac{c - c \cdot s(d_i(t))}{c - 1} \cdot p_\eta(t) - \frac{c - c \cdot s(d - d_i(t))}{c - 1} \cdot p_\eta(1-t) \right)$$

$$p_\eta(t) \propto p_\epsilon((x_j - x_i)t) = \frac{p_\gamma(\|(x_j - x_i)t\|)}{S_n(\|(x_j - x_i)t\|)} = \frac{p_\gamma(\|\epsilon_i\|)}{S_n(\|\epsilon_i\|)}$$

$$p_\eta(1-t) \propto p_\epsilon((x_i - x_j)(1-t)) = \frac{p_\gamma(\|(x_j - x_i)(1-t)\|)}{S_n(\|(x_j - x_i)(1-t))\|)} = \frac{p_\gamma(\|\epsilon_j\|)}{S_n(\|\epsilon_j\|)}$$

**Case $t \in [0, 1/2)$:**

$$\frac{p_\eta(t)}{p_\eta(t) + p_\eta(1-t)} = \frac{\dfrac{p_\gamma(\|\epsilon_i\|)}{S_n(\|\epsilon_i\|)}}{\dfrac{p_\gamma(\|\epsilon_i\|)}{S_n(\|\epsilon_i\|)} + \dfrac{p_\gamma(\|\epsilon_j\|)}{S_n(\|\epsilon_j\|)}} = \frac{1}{1 + \dfrac{p_\gamma(\|\epsilon_j\|)}{p_\gamma(\|\epsilon_i\|)} \cdot \dfrac{S_n(\|\epsilon_i\|)}{S_n(\|\epsilon_j\|)}} \ge \frac{1}{1 + \left( \dfrac{\|\epsilon_i\|}{\|\epsilon_j\|} \right)^{n-1}}$$

The last inequality comes from fact that $p_\gamma(\cdot)$ is a monotonically decreasing function and $\|\epsilon_i\| = \|(x_j - x_i)t\| < d/2 < \|(x_i - x_j)(1-t)\| = \|\epsilon_j\|$.

As $n$ grows, $(\|\epsilon_i\|/\|\epsilon_j\|)^{n-1}$ converges to 0 for any $t \in [0, 1/2)$. In other terms, the high-dimensionality of the input space essentially makes $(|\epsilon_i|/|\epsilon_j|)^{n-1}$ to be practically zero. As an example, in the CIFAR-10/100 benchmarks where the dimensionality $n$ equals $32 \cdot 32 \cdot 3 = 3071$, $t \in [0, 1/2 - 10^{-3}]$ gives $(|\epsilon_i|/|\epsilon_j|)^{n-1} < 4.6 \times 10^{-6}$. Essentially, we can regard $(|\epsilon_i|/|\epsilon_j|)^{n-1}$ as virtually zero in real-world scenarios, unless we encounter the unlikely cases where $t$ is extraordinarily near to $1/2$. Formally,

$$1 \ge \frac{p_\eta(t)}{p_\eta(t) + p_\eta(1-t)} \ge \frac{1}{1 + (\|\epsilon_i\|/\|\epsilon_j\|)^{n-1}} \approx 1 \longrightarrow \frac{p_\eta(t)}{p_\eta(t) + p_\eta(1-t)} \approx 1.$$

Accordingly, $\dfrac{p_\eta(1-t)}{p_\eta(t) + p_\eta(1-t)} \approx 0$.

$$g_{ij}(x(t))^* - g_{ij}(x(t))^*_{LS}$$
$$= \frac{1}{p_\eta(t) + p_\eta(1-t)} \cdot \left( \frac{c \cdot s(d_i(t)) - 1}{c - 1} \cdot p_\eta(t) - \frac{c \cdot s(d - d_i(t)) - 1}{c - 1} \cdot p_\eta(1-t) \right)$$

$$\approx \frac{c \cdot s(d_i(t)) - 1}{c - 1} \ge 0 \qquad (\because 1/c \le s(d_i(t)) \le 1)$$

$$g_{ij}(x(0))^*_{LS} = \frac{p_\eta(0) - p_\eta(1)}{p_\eta(0) + p_\eta(1)} \ge 0.$$

Using Lemma 1 and $g_{ij}(x(0))^*_{LS} \ge 0$, we have $g_{ij}(x(t))^*_{LS} \ge 0$ for $t \in [0, 1/2)$.

**Case $t \in (1/2, 1]$:**

Using Lemma 1 and $0 \le g_{ij}(x(t))^*_{LS} \le g_{ij}(x(t))^*$ for $t \in [0, 1/2)$, $0 \ge g_{ij}(x(t))^*_{LS} \ge g_{ij}(x(t))^*$ for $t \in (1/2, 1]$.

**Case** $t = 1/2$**:**

Using Lemma 1, $g_{ij}(x(1/2))^*_{LS} = g_{ij}(x(1/2))^* = 0$. $\qquad\square$

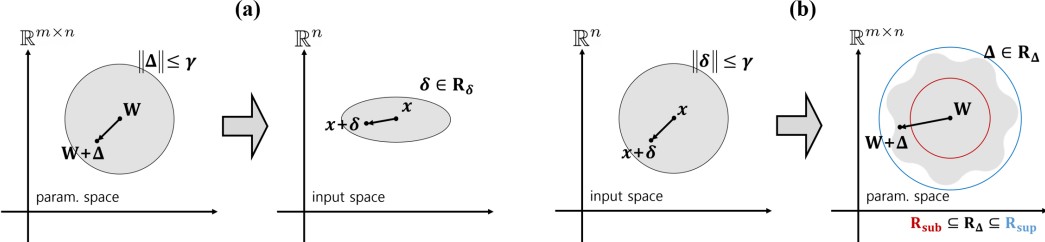

Figure 4: Illustrations of theorems on perturbation conversions. **(a)** The parameter perturbation region $\{\|\Delta\| \leq \gamma\}$ can be connected to an ellipsoidal perturbation region $R_\delta$ (Theorem 2.) **(b)** The input perturbation region $\{\|\delta\| \leq \gamma\}$ can be connected to parameter perturbation region $R_\Delta$. $R_\Delta$ has the subset $R_{sub} := \{\|\Delta\| \leq (\lambda_{min}/\|x_{max}\|)^2\}$ (Theorem 3) and the superset $R_{sup} := \{\|\Delta\| \leq \rho^2\}$ (Theorem 4.)

## C  CONVERTING PERTURBATIONS IN PARAMETER SPACE TO INPUT SPACE

Given weights $W \in \mathbb{R}^{m \times n}, b \in \mathbb{R}^m$, input $x \in \mathbb{R}^n$, and parameter perturbation region $\|\Delta\| \leq \gamma$, we want to find the region $R_\delta$ so that $\forall \|\Delta\| \leq \gamma, \exists \delta \in R_\delta$ $s.t.$ $\sigma(W(x+\delta)+b) = \sigma((W+\Delta)x+b)$ and $\forall \delta \in R_\delta, \exists \|\Delta\| \leq \gamma$ $s.t.$ $\sigma(W(x+\delta)+b) = \sigma((W+\Delta)x+b)$. In other words, we want to find the region $R_\delta$ so that for every element $e_1$ in region $\{\Delta \in \mathbb{R}^{m \times n} \mid \|\Delta\| \leq \gamma\}$ there exists an element $e_2$ in region $R_\delta$ satisfying the equation and vice versa.

Since $\sigma(\cdot) : \mathbb{R}^m \to (0,1)^m$ is a bijective function, $\sigma(W(x+\delta)+b) = \sigma((W+\Delta)x+b)$ $\iff W(x+\delta)+b = (W+\Delta)x+b$. This equality can be reduced to $W\delta = \Delta x$.

We will first examine the range of $\Delta x$ in the output space, given $\|\Delta\| \leq \gamma$. $\Delta x$ can be written in several ways:

$$\Delta x = \begin{bmatrix} | & | & & | \\ c_1 & c_2 & \cdots & c_n \\ | & | & & | \end{bmatrix} \begin{bmatrix} x_1 \\ x_2 \\ \vdots \\ x_n \end{bmatrix} = \begin{bmatrix} c_{11} & c_{12} & \cdots & c_{1n} \\ c_{21} & c_{22} & \cdots & c_{2n} \\ \vdots & \vdots & & \vdots \\ c_{m1} & c_{m2} & \cdots & c_{mn} \end{bmatrix} \begin{bmatrix} x_1 \\ x_2 \\ \vdots \\ x_n \end{bmatrix}$$

$$= c_1 x_1 + c_2 x_2 + \cdots + c_n x_n = \begin{bmatrix} c_{11} \\ c_{21} \\ \cdots \\ c_{m1} \end{bmatrix} x_1 + \begin{bmatrix} c_{12} \\ c_{22} \\ \cdots \\ c_{m2} \end{bmatrix} x_2 + \cdots + \begin{bmatrix} c_{1n} \\ c_{2n} \\ \cdots \\ c_{mn} \end{bmatrix} x_n$$

, where $c_i$ is the $i$th column vector and $c_{ij}$ is an element in $i$th row, $j$th column of $\Delta$.

Next, we will rewrite $\|\Delta\| \leq \gamma$ as the following constraints:

$\|\Delta\| \leq \gamma$
$\iff \sum_{i=1}^{m} \sum_{j=1}^{n} c_{ij}^2 \leq \gamma^2$
$\iff \sum_{j=1}^{n} \|c_j\|^2 \leq \gamma_j^2$ subject to $\gamma_1^2 + \gamma_2^2 + \cdots + \gamma_n^2 = \gamma^2$.

When we reexamine the above formulas in $\mathbb{R}^m$, finding the range of $\Delta x$ can be regarded as finding the range of linear combination of column vectors in $\mathbb{R}^m$ such that each column vector $c_i$ is restricted to $\|c_i\| \leq \gamma_i$.

Given two vectors $v_1$ and $v_2$ s.t. $\|v_1\| \leq \gamma_1$ and $\|v_2\| \leq \gamma_2$, $\|v_1 + v_2\| \leq \gamma_1 + \gamma_2$ . Trivially, for any $\alpha \in \mathbb{R}$, $\|\alpha \cdot v_1\| \leq |\alpha|\gamma_1$. That is, the range of linear combination $\Delta x = c_1 x_1 + c_2 x_2 + \cdots + c_n x_n$ is also a ball, i.e. $\|\Delta x\| \leq \sum_{i=1}^{n} |x_i|\gamma_i$ subject to $\sum_{i=1}^{n} \gamma_i^2 = \gamma^2$.

Finding the range of $\|\Delta x\|$ is now equivalent to finding the maximum radius of $\sum_{i=1}^{n} |x_i|\gamma_i$ with the constraint $\sum_{i=1}^{n} \gamma_i^2 = \gamma^2$. Using Lagrange multipliers method, let $r := [\gamma_1, \gamma_2, \cdots, \gamma_n]$, $f(r) := \sum_{i=1}^{n} |x_i|\gamma_i$, $g(r) := \sum_{i=1}^{n} \gamma_i^2 - \gamma^2$, and $L(r, \lambda) := f(r) - \lambda(g(r))$.

$$\frac{\partial L}{\partial \gamma_i} = |x_i| - 2\lambda\gamma_i = 0 \iff \gamma_i = \frac{|x_i|}{2\lambda}$$

Substituting the above equality to $g(r) = 0$,

$$\sum_{i=1}^{n} \frac{x_i^2}{4\lambda^2} - \gamma^2 = 0 \iff \lambda = \frac{\sqrt{\sum x_i^2}}{2\gamma}$$

$$\gamma_i = \frac{|x_i|}{2\lambda} = \frac{|x_i|\gamma}{\sqrt{\sum x_i^2}}$$

$$f(r) = \sum_{i=1}^{n} \frac{x_i^2 \gamma}{\sqrt{\sum x_i^2}} = \frac{\sum_{i=1}^{n} x_i^2}{\sqrt{\sum_{i=1}^{n} x_i^2}} \gamma = \|x\| \cdot \gamma$$

Therefore, $\|\Delta x\| \leq \|x\|\gamma$.

We now consider the LHS of equation $W\delta = \Delta x$. Let $W = U\Sigma V^\top$ be the SVD Decomposition of $W \in \mathbb{R}^{m \times n}$. Multiplying $U^\top$ to both sides of the equation, $\Sigma V^\top \delta = U^\top \Delta x$. The inequality induced by $L_2$ norm, i.e. ball, does not change when we multiply any orthogonal matrix. Thus, $\|U^\top \Delta x\| \leq \|x\|\gamma$.

Let $\delta' := V^\top \delta = [\delta_1', \cdots, \delta_n']^\top$.

$$\Sigma V^\top \delta = \Sigma \delta' = \begin{bmatrix} \sigma_1 & & & 0 & \cdots & 0 \\ & \ddots & & \vdots & & \vdots \\ & & \sigma_m & 0 & \cdots & 0 \end{bmatrix} \begin{bmatrix} \delta_1' \\ \vdots \\ \delta_m' \\ \delta_{m+1}' \\ \vdots \\ \delta_n \end{bmatrix} = \begin{bmatrix} \sigma_1 \delta_1' \\ \vdots \\ \sigma_m \delta_m' \end{bmatrix}$$

Since $\|U^\top \Delta x\| \leq \|x\|\gamma$ and $\Sigma \delta' = U^\top \Delta x$, $\|\Sigma \delta'\| \leq \|x\|\gamma$, i.e.

$$\sigma_1^2 \delta_1'^2 + \cdots + \sigma_m^2 \delta_m'^2 + 0 \cdot (\sigma_{m+1}^2 \delta_{m+1}'^2 + \cdots + \sigma_n^2 \delta_n'^2) \leq \|x\|^2 \gamma^2$$

However since $0 \cdot (\sigma_{m+1}^2 \delta_{m+1}'^2 + \cdots + \sigma_n^2 \delta_n'^2) = 0$ holds for any $\delta$, i.e. the general solution to $Wa = Wb$ where $a \neq b$, we need not contain it in our perturbation region $R_\delta$ which is induced by $\|\Delta\| \leq \gamma$. Then, the above inequality represents a $m$-dim region bounded by a $m$-dim ellipsoid whose principal semi-axes have lengths $(\sigma_1 \|x\|\gamma)^{-1}, \cdots, (\sigma_n \|x\|\gamma)^{-1}$ with respect to $\delta' \in \mathbb{R}^n$. Subsequently, the region of interest $R_\delta \in \mathbb{R}^n$ is an rotated $m$-dim ellipsoid whose principal semi-axes have lengths $(\sigma_1 \|x\|\gamma)^{-1}, \cdots, (\sigma_n \|x\|\gamma)^{-1}$ with respect to $\delta \in \mathbb{R}^n$.

# D    CONVERTING PERTURBATIONS IN INPUT SPACE TO PARAMETER SPACE

Given weights $W \in \mathbb{R}^{m \times n}$, input $x \in \mathbb{R}^n$, and parameter perturbation region $\|\delta\| \leq \gamma$, we want to find the region $R_\Delta$ so that $\forall \|\delta\| \leq \gamma, \exists \Delta \in R_\Delta \ s.t. \ W\delta = \Delta x$ and $\forall \Delta \in R_\Delta, \exists \|\delta\| \leq \gamma \ s.t. \ W\delta = \Delta x$.

Using SVD decomposition, $W = U\Sigma V^\top$, where $\Sigma$ is a diagonal matrix with entries $\sigma_1, \cdots, \sigma_n$.

$W\delta = U\Sigma V^\top \delta = U\Sigma \delta'$, where $\delta' := V^\top \delta$. Since rotating or reflecting does not change the region of a ball, $\|\delta\| \leq \gamma$ gives $\|\delta'\| \leq \gamma$, i.e. $\delta_1'^2 + \cdots \delta_n'^2 \leq \gamma^2$.

Let $\delta'' := [\delta_1'', \cdots, \delta_m''] = \Sigma \delta' = [\sigma_1 \delta_1', \cdots, \sigma_m \delta_m']$. $\forall i \in [m], \sigma_i^{-1} \delta_i'' = \delta_i'$. Then,

$$\frac{\delta_1''^2}{\sigma_1^2} + \cdots + \frac{\delta_m''^2}{\sigma_m^2} \leq \gamma^2 - \left(\delta_{m+1}'^2 + \cdots \delta_n'^2\right) \tag{1}$$

The maximum value of RHS in eq. 1 is $\gamma^2$, when $\left(\delta_{m+1}'^2 + \cdots \delta_n'^2\right) = 0$. This indicates that $\delta''$ resides within an ellipsoid with with principle semi-axes of lengths $\lambda_i := \sigma_i \gamma, i \in [m]$. Thus, $U\delta'' = W\delta$ is a region bounded by an rotated ellipsoid.

Now, we will examine the region $R_\Delta$ such that $\Delta x \ (\Delta \in R_\Delta)$ forms a rotated ellipsoid with principle semi-axes of lengths $\lambda_i$. Unlike the case of converting parameter space's perturbation region to input space's in Appendix B, $R_\Delta$ need not be in a form of ellipsoid. Instead, we provide a superset $R_{sup}$ and a subset $R_{sub}$ of $R_\Delta$ in the form of a ball such that $R_{sub} \subseteq R_\Delta \subseteq R_{sup}$.

Let $W$ be deomposed into $U\Sigma V^\top$ using SVD decomposition. For now, we will consider the special case of $W$ where $U = I$, i.e. the region of $W\delta$ is bounded by an ellipsoid alligned with standard basis. Afterwards, we will consider the general case of $W$, i.e. the region of $W\delta$ is bounded by a rotated ellipsoid.

Let $d_{ij}$ denote the $i$th row, $j$th column element of $\Delta \in \mathbb{R}^{m \times n}$ and $x_i$ the $i$th element of $x \in \mathbb{R}^n$. Since the range of $\Delta x$ is an ellipsoid, $\Delta x$ must satisfy the ellipsoid inequality

$$\frac{(x_1 d_{11} + x_2 d_{12} + \cdots + x_n d_{1n})^2}{\lambda_1^2} + \cdots + \frac{(x_1 d_{m1} + x_2 d_{m2} + \cdots + x_n d_{1n})^2}{\lambda_m^2} \leq 1$$

Let $r_i$ denote the $i$th row vector of $\Delta$, and let $X$ denote $xx^\top$. The above inequality can be rewritten as:

$$\frac{r_1^\top X r_1}{\lambda_1^2} + \frac{r_2^\top X r_2}{\lambda_2^2} + \cdots + \frac{r_m^\top X r_m}{\lambda_m^2} \leq 1 \tag{2}$$

Since we are interested in finding the region of $\Delta$ in $\mathbb{R}^{m \times n}$ space, we may think of it as a vector $d = [r_1^\top, r_2^\top, \cdots, r_m^\top]$ in $\mathbb{R}^{(m \times n)}$ rather than as a matrix. Then, inequation 2 can be rewritten as:

$$d^\top X_\lambda d \leq 1, \text{ where } X_\lambda := \begin{bmatrix} X/\lambda_1^2 & & & \\ & X/\lambda_2^2 & & \\ & & \ddots & \\ & & & X/\lambda_m^2 \end{bmatrix} \in \mathbb{R}^{(m \times n)^2}$$

One property of $X_\lambda$ is that it is a rank $m$ matrix with singular values $\|x\|^2/\lambda_1^2, \cdots, \|x\|^2/\lambda_m^2$, regarding that $X/\lambda_i^2$ is a rank 1 matrix with singular value $\|x\|^2/\lambda_i^2$. Another property is that $X_\lambda$ is a positive-semidefinite matrix $(\because \forall i \in [m], \|x\|^2/\lambda_i^2 \geq 0.)$

When we think of a single input $x$, the area of $d$ satisfying $d^\top X_\lambda d \leq 1$ is not bounded. However, when we consider the constraint over multiple values of input datapoints $\{x_1, x_2, \cdots, x_N\}(N \gg n)$ that spans $\mathbb{R}^n$, the area becomes bounded. One justification of the multiple constraints is that when we consider $x$ a uniform random variable over the input datapoints, the region of $d$ that satisfies all the possible constraint is $\cup_{i=1}^{N} d^\top X_\lambda^{(i)} d \leq 1$, where $X_\lambda^{(i)}$ denotes $X_\lambda$ for $x = x_i$. Another justification is

that when we reach a local plateau in training parameter $W$, there is little or no change in the value of $W$.

The following lemma and theorems provide a subset $R_{sub}$ and superset $R_{sup}$ of $R_\Delta$ in the form of balls in the parameter space.

**Lemma 3.** *Let $R$ be the region of $x \in \mathbb{R}^n$ satisfying the inequality $x^\top A x \leq 1$, where $A$ is a non-zero positive semi-definite matrix having $\sigma_{max}$ as the maximum nonzero singular value. Let $R'$ be the region of $x \in \mathbb{R}^n$ satisfying the ineqaulity $x^\top x \leq \sigma_{max}^{-1}$. $R \subseteq R'$.*

*Proof.* We handle two cases where $rank(A) = m$ and $rank(A) < m$.

**Case $rank(A) = m$:**

Using SVD Decomposition, $A = U\Sigma U^\top$, where $\Sigma = \begin{bmatrix} \sigma_1 & & \\ & \ddots & \\ & & \sigma_n \end{bmatrix}$

$x^\top A x = x^\top U \Sigma U^\top x = x'^\top \Sigma x' \leq 1$, where $x' := U^\top x$

Let $x'$ be represented as $x' = [x'_1, \cdots, x'_n]$.

The constraint induced by $R$ can be rewritten as:

$$x'^\top \Sigma x' = \sigma_1 x_1'^2 + \cdots + \sigma_n x_n'^2 \leq 1, \text{ where } \Sigma = U^\top A U$$

Let $x \in \mathbb{R}^n$ be some vector satisfyig $x^\top x \leq \sigma_{max}$. Since $U$ is an orthogonal matrix and $x^\top x \leq \sigma_{max}^{-1}$ is an equidistant ball that is invariant under rotations and reflections, the constraint induced by $R'$ can be rewritten as $x'^\top x' \leq \sigma_{max}^{-1}$, where $x' = U^\top x$.

To prove $x \in R'$ implies $x \in R$, we will show $x'^\top x' \leq \sigma_{max}^{-1}$ implies $x'^\top \Sigma x' \leq 1$.

$$x'^\top x' \leq \sigma_{max}^{-1} \iff \sigma_{max} x'^\top x' \leq 1$$

Let $\epsilon_i := \sigma_{max} - \sigma_i$. Then, $\forall i \in [n], \epsilon_i \geq 0$.

$$
\begin{aligned}
\sigma_{max} x'^\top x' - \sum_{i=1}^n \epsilon(x'_i)^2 &\leq 1 - \sum_{i=1}^n \epsilon_i (x'_i)^2 && (\because \sigma_{max} x'^\top x' \leq 1) \\
&\leq 1 && (\because \forall i \in [n], \epsilon_i (x'_i)^2 \geq 0)
\end{aligned}
$$

**Case $rank(A) < m$:**

Let $rank(A) = k < m$. $A$ can be represented as $U\Sigma U^\top$ using SVD decomposition, where $\Sigma$ is a diagonal matrix whose first $k$ elements are non-zero singular values $\sigma_1, \cdots, \sigma_k$.

$$x^\top A x = x^\top U \Sigma U^\top x = x' \Sigma x' \leq 1, \text{ where } \Sigma = U^\top A U \text{ and } x' := U^\top x$$

Let $x'$ be represented as $[x'_1, \cdots, x'_n]$. The constrained induced by $R$ can be rewritten as:

$$x'^\top \Sigma x' = \sigma_1 x_1'^2 + \cdots + \sigma_k x_k'^2 \leq 1$$

Let $x \in \mathbb{R}^n$ be any vector satisfying $x^\top x \leq \sigma_{max}^{-1}$. Since ball is equidistant, $x^\top x \leq \sigma_{max}^{-1} \iff x''^\top x' \leq \sigma_{max}^{-1}$, where $x' = U^\top x$.

To prove $x \in R'$ implies $x \in R$, we will show $x'^\top x' \leq \sigma_{max}^{-1}$ implies $x' \Sigma x' \leq 1$.

$$x'^\top x' \leq \sigma_{max}^{-1} \iff \sigma_{max} x'^\top x' \leq 1 \iff \sum_{i=1}^n \sigma_{max} (x'_i)^2 \leq 1$$

Let $\epsilon_i := \sigma_{max} - \sigma_i$. Then, $\forall i \in [n], \epsilon_i \geq 0$.

$$\sum_{i=1}^{k}(\sigma_{max} - \epsilon_i)x_i'^2 \leq \sigma_{max}x^\top x - \sum_{i=1}^{k}\epsilon_i(x_i')^2 \qquad \left(\because \sum_{i=k+1}^{n}\sigma_{max}x_i^2 \geq 0\right)$$

$$\leq 1 - \sum_{i=1}^{k}\epsilon_i(x_i')^2 \qquad \left(\because \sigma_{max}x'^\top x' \leq 1\right)$$

$$\leq 1 \qquad \left(\because \forall i \in [k], \epsilon_i(x_i')^2 \geq 0\right)$$

Since $\sum_{i=1}^{k}(\sigma_{max} - \epsilon_i)x_i'^2 = x'^\top \Sigma x', \; x'^\top \Sigma x' \leq 1$. $\qquad \square$

**Theorem 3.** *Given $W \in \mathbb{R}^{m \times n}$, $D = \{x_1, \cdots, x_N\}(x_i \in \mathbb{R}^n/\{0\}$ for $i \in [N])$, and input perturbation region $\{\delta \in \mathbb{R}^n \mid \|\delta\| \leq \gamma\}$, let $x_{max} := \arg\max_{x_i}\|x_i\|$ and $\lambda_{min} := \min\{\lambda_1, \cdots, \lambda_m\}$. Then, $\{\Delta \in \mathbb{R}^{m \times n} \mid \|\Delta\| \leq (\|x_{max}\|^2/\lambda_{min}^2)^{-1}\}$ is the subset of $R_\Delta$.*

*Proof.* We will rewrite theorem 3 as the following statement:

*Given a set of datapoints $D = \{x_1, x_2, \cdots, x_N\}(x_i \in \mathbb{R}^n/\{0\}, i \in [N])$, let $R$ be the region of $d \in \mathbb{R}^{m \times n}$ satisfying the inequality $d^\top X_\lambda d \leq 1$ for all $x \in D$. Let $R'$ be the region of $d \in \mathbb{R}^{m \times n}$ satisfying $d^\top d \leq (\|x_{max}\|^2/\lambda_{min}^2)^{-1}$, where $x_{max} := \arg\max_{x_i}\|x_i\|$ and $\lambda_{min} := \min\{\lambda_1, \cdots, \lambda_m\}$. $R' \subseteq R$.*

Remark that $X_\lambda^{(i)} = \begin{bmatrix} x_i^\top x_i/\lambda_1^2 & & & \\ & x_i^\top x_i/\lambda_2^2 & & \\ & & \ddots & \\ & & & x_i^\top x_i/\lambda_m^2 \end{bmatrix}$. $X_\lambda^{(i)}$ is a rank $m$ matrix with

singular values $\|x_i\|^2/\lambda_1^2, \cdots, \|x_i\|^2/\lambda_m^2$.

Let $R_i$ denote the region of $d \in \mathbb{R}^n$ satisfying $d^\top X_\lambda^{(i)}d \leq 1$, and let $R_i'$ denote the region $d^\top d \leq \left(\frac{\|x_i\|^2}{\lambda_{min}^2}\right)^{-1} \cdot \frac{\|x_i\|^2}{\lambda_{min}^2}$ being the largest singular value of $X_\lambda^{(i)}$, $R_i' \subseteq R_i$ by Lemma 1. Since this holds for all $i \in [N]$, $\bigcup_{i=1}^{N} R_i' \subseteq \bigcup_{i=1}^{N} R_i$. $\bigcup_{i=1}^{N} R_i = R$, and $\bigcup_{i=1}^{N} R_i' = R'$ is a ball with smallest radius, i.e. $d^\top d \leq \left(\frac{\|x_{max}\|^2}{\lambda_{min}^2)^{-1}}\right)$. $\qquad \square$

**Theorem 4.** *Given $W \in \mathbb{R}^{m \times n}$, $D = \{x_1, \cdots, x_N\}(x_i \in \mathbb{R}^n/\{0\}$ for $i \in [N])$, and input perturbation region $\{\delta \in \mathbb{R}^n \mid \|\delta\| \leq \gamma\}$, let $R_i := \{d \in \mathbb{R}^{m \times n} \mid d^\top X_\lambda^{(i)}d \leq 1\}$ and $\Gamma := \{R_i \mid i \in [N]\}$. Then, $\{\arg\min_{R_1, \cdots, R_n \in \Gamma} \max_{\rho \in \cup_{i \in [n]} R_i} \|\rho\|^2\}$ is the superset of $R_\Delta$.*

*Proof.* Let $R_1^*, \cdots, R_n^*$ denote the elements of $\Gamma$ satisfying $\arg\min_{R_1, \cdots, R_n \in \Gamma} \max_{\rho \in \cup R_i} \|\rho\|^2$.

$R = \bigcup_{i=1}^{N} R_i \subseteq \bigcup_{i=1}^{n} R_i^* \subseteq \max_{\rho \in \cup R_i} \|\rho\|^2$. $\qquad \square$

We have so far addressed the case where $U = I$ for $W = U\Sigma V^\top$ in the equation $W\delta = \Delta x$. Now, let us consider the general case of full rank matrix $W$.

$\Delta \in \mathbb{R}^{m \times n}$ can be represented as either column vectors $[c_1, c_2, \cdots, c_n]$ or row vectors $[r_1, r_2, \cdots, r_m]^\top$. The equation $W\delta = \Delta x$ can be rewritten as:

$$\Sigma V^\top \delta = U^\top \Delta x = U^\top[c_1, c_2, \cdots, c_n]x = [U^\top c_1, U^\top c_2, \cdots, U^\top c_n]x$$

Let $\Delta' := U^\top\Delta = [c'_1, c'_2, \cdots, c'_n] = [r'_1, r'_2, \cdots, r'_m]^\top$, and let $d'$ be the flattened vector representation $[r'^\top_1, r'^\top_2, \cdots, r'^\top_m]$ of $\Delta'$. Then, finding $R_\Delta$ is equivalent to finding the region of $\Delta'$ satisfying $d'^\top X_\lambda d' \leq 1$ and multiplying $U$ to $\Delta'$.

The relationship between $\Delta'$ and $\Delta$ can be expressed as:

$$U_{diag}\begin{bmatrix} c'_1 \\ c'_2 \\ \vdots \\ c'_n \end{bmatrix} = \begin{bmatrix} c_1 \\ c_2 \\ \vdots \\ c_n \end{bmatrix}, \text{ where } U_{diag} := \begin{bmatrix} U & & & \\ & U & & \\ & & \ddots & \\ & & & U \end{bmatrix} \in \mathbb{R}^{(m\times n)^2}$$

$U_{diag}$ is an orthogonal matrix since $U$ is an orthogonal matrix. Furthermore, any permutation $\pi$ that permutes the row vectors of $U_{diag}$ also results in another orthogonal matrix $U^\pi_{diag}$. Then for some $\pi$, $U^\pi_{diag}[r'^\top_1, r'^\top_2, \cdots, r'^\top_m]^\top = [r^\top_1, r^\top_2, \cdots, r^\top_m]^\top$, i.e. $U^\pi_{diag}d' = d$. Since the region of a ball is not affected by rotations or reflections, the superset and the subset obtained in Theorem 1 and 2 are not affected. In other words,

$$R_\Delta = \{d \in \mathbb{R}^{m\times n} \mid \forall i \in [n], d^\top X^{(i)}_\lambda d \leq 1\}$$

$$R_{sub} = \{d \in \mathbb{R}^{m\times n} \mid d^\top d \leq \left(\frac{\|x_{max}\|^2}{\lambda^2_{min}}\right)^{-1}\}$$

$$R_{sup} = \{d \in \mathbb{R}^{m\times n} \mid \underset{R_1,\cdots,R_n\in\Gamma}{\arg\min}\ \underset{\rho\in\cup R_i}{\max}\ \|\rho\|^2\}$$

satisfies $R_{sub} \subseteq R_\Delta \subseteq R_{sup}$.

Lastly, we provide the implications of our theorem:

**Definition 3.** (b-flat local minima) Given any real-valued loss function $L$ and dataset $D = \{(x_1, y_1), (x_2, y_2), \cdots, (x_N, y_N)\}$, a model parameter $\theta$ is said to have $b$-flat minima if the following conditions hold:

i) $\forall\|\epsilon\| \leq b, \mathbb{E}_{(x,y)\sim D}[L(f(x;\theta), y)] = \mathbb{E}_{(x,y)\sim D}[L(f(x;\theta + \epsilon), y)]$

ii) $\forall\|\epsilon\| > b, \mathbb{E}_{(x,y)\sim D}[L(f(x;\theta), y)] < \mathbb{E}_{(x,y)\sim D}[L(f(x;\theta + \epsilon), y)]$

Given any real-valued loss $L$ and dataset $D$, let $\theta^*$ denote any optimal parameter that minimizes loss w.r.t. dataset, i.e. $\theta^* := \arg\min_\theta \mathbb{E}_{(x,y)\sim D}[L(f(x;\theta), y)]$. Analogously, let $\theta^*_\gamma$ denote any optimal parameter such that $\forall\|\delta\| \leq \gamma, \theta^*_\gamma := \arg\min_\theta \mathbb{E}_{(x,y)\sim D}[L(f(x + \delta;\theta), y)]$.

The following holds for any linear classifier $f : x \mapsto \sigma(Wx + b)$:

**Proposition 1.** *$\theta^*$ can have 0-flat minima.*

*Proof.* There exists $\theta^*$ such that $\forall(x, y) \in D, \theta^* := \arg\min_\theta \mathbb{E}_{(x,y)\sim D}[L(f(x;\theta), y)]$ and $\forall\delta \in \mathbb{R}^m/\{0\}, L(f(x + \delta;\theta), y) > L(f(x;\theta), y)$. □

**Corollary 1.** *$\theta^*_\gamma$ has $b$-flat minima with $b \geq (\|x_{max}\|^2/\lambda^2_{min})^{-1}$.*

*Proof.* For $\theta^*_\gamma$ to be an optimal parameter, $\forall(x, y) \in D, \|\delta\| \leq \gamma, L(f(x + \delta; \theta^*_\gamma), y) = L(f(x;\theta^*_\gamma), y) = \min_\theta L(f(x;\theta), y)$. Using results from Theorem 3, $\forall(x, y) \in D, \|\Delta\| \leq (\|x_{max}\|^2/\lambda^2_{min})^{-1}, L(f(x;\theta^*_\gamma + \Delta), y) = L(f(x;\theta^*_\gamma), y) = \min_\theta L(f(x;\theta), y)$. □

The proposition and the collorary implies that exploiting perturbation-based algorithm will provide higher lower bound of $b$ $((\|x_{max}\|^2/\lambda^2_{min})^{-1} > 0)$ for the $b$-flat minima of the optimal parameter.

# E ADDITIONAL EXPERIMENTAL DETAILS

In this section, we provide an in-depth discussion on the experiments conducted in the main paper, as well as present additional experimental findings related to SPIDER including hyperparameter sensitivity analysis.

## E.1 EXPERIMENTAL DETAILS

In our experiments, the hardware resources employed differ based on the complexity of the tasks. All the tasks in the main paper related to CIFAR-10/100, and Tiny-ImageNet has been handled using 8 NVIDIA RTX A5000 GPUs. While a single A5000 GPU could have sufficed for the evaluation, the multi-GPU setup have been opted for to facilitate the extensive evaluations for each baseline methods. ImageNet training experiment has been carried out utilizing a single A100 GPU, owing to its superior computational capacity.

### E.1.1 MAIN TABLE

**Robustness Against General Corruptions**

For CIFAR-10/100 experiments, we use WRN-40-2 architecture exploited in (Hendrycks et al., 2021b). For Tiny-ImageNet and ImageNet experiment, we use ResNet18 (He et al., 2015) as our backbone. SGD with momentum value of 0.9 has been used in all our experiments. Cosine learning rate decay scheduling (Loshchilov & Hutter, 2017) with initial learning rate of 0.1, 0.01, and 0.01 has been used respectively for CIFAR-10/100, Tiny-ImageNet, and ImageNet experiments to train a model until convergence. Models have been trained for 400, 100, and 90 epochs for CIFAR-10/100, Tiny-ImageNet, and ImageNet benchmarks respectively.

The search space for hyperparameters $(\tau, \xi)$ introduced by SPIDER instantiation are as follows. For $\tau$, the range was set to $[0.0, 20.0]$ for CIFAR-10/100 and $[0.0, 30.0]$ for Tiny-ImageNet. $\xi$ was tested within the range of $[0, 1 - c^{-1}]$, with $c$ representing the number of classes in the given dataset. The rescaling and clipping algorithm by Rauber (Rauber & Bethge, 2020) was utilized to keep perturbed data points within a valid domain (i.e. $[0, 1]^m$).

**Evaluation of Robustness to Common Data Corruptions**

**Benchmark Statistics:** CIFAR-10/100-C, Tiny-ImageNet-C, and ImageNet (Deng et al., 2009) datasets contain 15 distinct corruption types: brightness changes, contrast alterations, defocus blur, elastic transformations, fog addition, frost addition, Gaussian blur, glass distortion, impulse noise, jpeg compression, motion blur, pixelation, shot noise, snow addition, and zoom blur with 5 different severity levels per each corruption. mCE calculates the average error of a model across all the distinct corruptions and severity levels. For ImageNet, we report mCE with normalization suggested as in Hendrycks & Dietterich (2019).

**Evaluation Process:** During training and validation, a model is trained and validated on the uncorrupted training and validation data. The validation data has been constructed using 20% of the training data. The model achieving the best validation accuracy is chosen and evaluated on a corrupted dataset, where the corruption types were not encountered either at the training or the validation stage. To reduce the discrepancy between the clean, non-augmented dataset and the corrupted dataset, we augment the clean validation data of CIFAR, Tiny-ImageNet, and ImageNet datasets using augmentations from (Mintun et al., 2021) during the validation phase. These augmentations are distinct from the corruptions used in the common corruption benchmarks (CIFAR/100-C, Tiny-ImageNet-C, and ImageNet-C). In short, we train a model using SPIDER on clean data, select the models with the highest accuracy on the augmented validation data (using functions from (Mintun et al., 2021)), and then assess these models' robustness on common corruption benchmarks.

**Evaluation of Robustness to Adversarial Attacks**

To assess the model's robustness against adversarial attacks, we use untargeted PGD attacks based on $L_2$ and $L_\infty$ norms. We have chosen to use the absolute value $\alpha$ as the coefficient of gradient ascent for clarity, instead of using a relative step size with respect to $\epsilon$. For $L_2$ attacks, PGD-20 attack with $(\epsilon, \alpha) = (0.5, 1/800)$ has been used. For $L_\infty$ attacks, we have used PGD-7 attack with $(\epsilon, \alpha) = (8/255, 2/255)$ for CIFAR-10/100, PGD-3 attack with $(\epsilon, \alpha) = (3/255, 1/255)$ for

the Tiny-ImageNet experiment, and PGD-2 $L_\infty$ attack with $(\epsilon, \alpha) = (1/255, 1/510)$ for ImageNet experiment. Essentially, more intense attacks have been applied to simpler datasets, while milder attacks have been used for more complex datasets, with the CIFAR experiments' attack configurations borrowed from the (Yang et al., 2020). We then evaluate the adversarial robustness of models trained following the common corruption evaluation protocol detailed above. The key interest here is not to show that SPIDER is setting new records for robustness, but to demonstrate that SPIDER enhances both common corruption robustness and adversarial robustness compared to previous augmentation methods that had negligible impact on adversarial robustness.

### E.1.2   BOUNDARY THICKNESS

Our approach adheres closely to the original paper that introduced the boundary thickness metric (Yang et al., 2020). For each data point $x_i$ in the dataset, labeled with one-hot encoded label $i$, we generate a corresponding adversarial instance $x_j$. This is achieved by conducting an attack on $x_i$ targeting a randomly selected class $j$ that differs from $i$. We use an $L_2$ PGD-20 untargeted attack with parameters $\epsilon = 5.0$ and $\alpha = 1.0$ to produce these adversarial instances. The integral $\int_0^1 I\{\alpha < g_{ij}x(t) < \beta\}dt$ is approximated by dividing the segment into 128 data points and determining the fraction of points that fall within the interval from $\alpha$ to $\beta$. For the purpose of measuring the mean and standard deviations of the boundary thickness, we generated 1600 data points, constructed from 50 batches of 32 images each, along with their adversarial counterparts. The data in Table 1 was calculated using baseline methods and SPIDER in combination with these baselines, using the weights obtained from the main experiment. Table 2 was calculated training the models that had identical configurations as the previous models, except with the smoothing function removed.

### E.1.3   FLATNESS

Flatness is evaluated by sampling parameter fluctuations of growing radii and calculating the average loss on the model with the adjusted parameter. To elaborate, for every radius value, three independent and identical models, trained using either standalone baseline augmentation methods or a combination of SPIDER with the baselines, are utilized for flatness computation. For every model, 50 independent parameter perturbations are sampled and implemented (yielding a total of 150 disturbed weight samples) to determine the average loss related to the respective radius.

### E.1.4   ABLATION TABLE

The Baseline and SPIDER columns have statistics from the main table. The hyperparameter value $(\tau)$ of the removal of smoothing function (No LS) have been found following the same hyperparameter search space used in main table (Appendix E.1.1.) For standard label smoothing (STD LS), the degree of smoothing has been set as hyperparameter value and optimized using TPE sampler from Optuna library.

## E.2   ADDITIONAL EXPERIMENT

### E.2.1   HYPERPARAMETER SENSITIVITY

**Analysis of Figure 5:** The depicted charts illustrate the effect of perturbation sensitivity on the performance of the model trained with SPIDER augmentation solely. We gauged this sensitivity by keeping the shape of the exponential smoothing function constant - specifically, we set $s(\tau) = 0.5$ for the CIFAR-10/100 experiments, and observed the performance as the radius $\tau$ grew. As expected, an overly large perturbation radius risks pushing datapoints into the submanifolds of different labels, which degrades clean accuracy. Conversely, an excessively small perturbation radius fails to provide sufficient robustness enhancement, as evidenced by the elevated mCE and augmented clean error values for smaller radii. As the radius increases, there is an initial decrease in mCE, indicating increased robustness against corruptions. However, the trend begins to reverse for larger radii, with an accompanying rise in clean and augmented clean error. This pattern underscores the necessity of an appropriate balance in perturbation size to maintain performance across both clean and corrupted conditions. Despite these variations, the mean corruption error and augmented clean error consistently

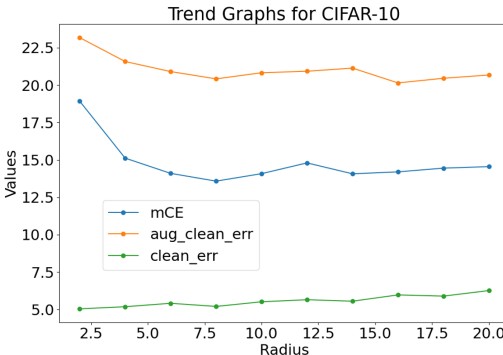 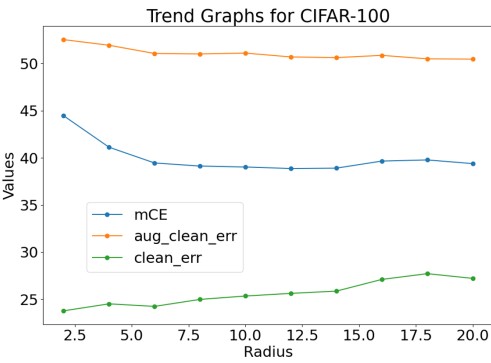

Figure 5: Sensitivity to increasing perturbations for CIFAR-10 and CIFAR-100. The plots depict the relationships between the radius of perturbation and the mean corruption error (mCE), augmented clean error (aug_clean_err), and clean error (clean_err) for the CIFAR-10 (left) and CIFAR-100 (right) datasets, with $\xi = 0.4$ and $0.49$ respectively. Each point represents the error rate obtained with a different radius of parameter perturbation.

stay beneath the baseline's performance, thereby suggesting that SPIDER improves model robustness compared to the baseline approach.

Table 7: Comparison of training times and performance between baselines with the addition of SPIDER. The use of Optuna's automated hyperparameter search algorithm with pruning alleviates the added training cost.

| Baselines | no aug. | AugMix | DeepAug. | PixMix |
|---|---|---|---|---|
| Pruned Training Duration(hour) | 14.84 | 14.06 | 39.51 | 27.98 |
| Predicted Full Training Duration (hour) | 58.15 | 71.99 | 171.54 | 140.90 |
| Speed Gain (x Faster) | 3.92 | 5.12 | 4.34 | 5.04 |
| Equivalent Non-Pruned Trials | 7.65 | 5.86 | 6.91 | 5.96 |
| Best Trial Index out of 30 | 2 | 3 | 2 | 5 |

**Analysis of Table 7:** The table compares the impact of incorporating SPIDER into various baseline augmentation methods in terms of training duration and performance. As the introduction of SPIDER brings additional hyperparameters for shaping the perturbation and smoothing functions, there is a potential for an increase in training time due to the associated hyperparameter search. To mitigate this, we have implemented Optuna's automated hyperparameter searching algorithm with an early stopping feature, also known as 'pruning'. The results show that the pruned training duration for each augmentation method (no augmentation, AugMix, DeepAugment, and PixMix) is significantly less than the predicted full training duration, indicating a substantial speed gain. The number of equivalent non-pruned trials ranges from approximately 5.86 to 7.65, implying that the use of pruning enables the same level of hyperparameter exploration to be achieved in a fraction of the time. The best trial index out of 30 shows that successful models can be identified relatively early in the process, further emphasizing the efficiency of the combined use of SPIDER with automated hyperparameter search and pruning. This approach, therefore, effectively alleviates the potentially increased training cost associated with the introduction of SPIDER's additional hyperparameters.

### E.2.2 VARYING BACKBONE NETWORKS

SPIDER demonstrates substantial improvement in robustness against common corruptions and adversarial attacks across various neural network architectures. This supports previous research, which found that augmentation techniques enhancing robustness retain their effectiveness and influence on model resilience, regardless of differences in the underlying network structures (Hendrycks et al., 2021b;a).

Table 8: Evaluation of robustness over different backbone networks on CIFAR-100 benchmark.

| Backbone | Method | mCE ↓ | | $L_2$ (PGD) ↓ | | $L_\infty$ (PGD) ↓ | |
|---|---|---|---|---|---|---|---|
| | | original | +SPIDER | original | +SPIDER | original | +SPIDER |
| DenseNet | no aug. | 57.67 ± 0.38 | 45.73 ± 0.56 | 99.44 ± 0.08 | 66.46 ± 0.38 | 100.00 ± 0.00 | 99.52 ± 0.06 |
| | AugMix | 41.79 ± 0.46 | 36.72 ± 0.15 | 96.94 ± 0.21 | 62.72 ± 0.43 | 99.98 ± 0.02 | 99.21 ± 0.09 |
| | DeepAug. | 44.92 ± 0.31 | 37.76 ± 0.29 | 97.37 ± 0.19 | 74.77 ± 0.26 | 89.05 ± 0.40 | 99.86 ± 0.01 |
| | PixMix | 35.71 ± 0.17 | 35.40 ± 0.07 | 99.97 ± 0.01 | 62.43 ± 0.19 | 99.99 ± 0.01 | 99.40 ± 0.05 |
| WRN-40-1 | no aug. | 57.07 ± 0.48 | 43.56 ± 0.21 | 99.15 ± 0.06 | 77.74 ± 0.55 | 99.99 ± 0.01 | 99.61 ± 0.03 |
| | AugMix | 43.82 ± 0.27 | 35.92 ± 0.06 | 97.38 ± 0.09 | 86.91 ± 0.12 | 99.99 ± 0.01 | 99.26 ± 0.13 |
| | DeepAug. | 44.38 ± 0.33 | 36.36 ± 0.13 | 96.10 ± 0.21 | 79.34 ± 0.52 | 99.98 ± 0.01 | 99.60 ± 0.06 |
| | PixMix | 37.76 ± 0.29 | 37.48 ± 0.06 | 92.69 ± 0.35 | 87.86 ± 0.39 | 99.95 ± 0.01 | 99.90 ± 0.03 |
| AllConvNet | no aug. | 56.71 ± 0.11 | 43.35 ± 0.22 | 92.40 ± 0.15 | 66.50 ± 0.33 | 99.98 ± 0.01 | 99.50 ± 0.08 |
| | AugMix | 42.52 ± 0.43 | 35.92 ± 0.06 | 85.49 ± 0.42 | 62.83 ± 0.35 | 99.93 ± 0.02 | 99.84 ± 0.01 |
| | DeepAug. | 42.39 ± 0.18 | 36.39 ± 0.35 | 88.66 ± 0.31 | 74.84 ± 0.24 | 99.87 ± 0.02 | 99.86 ± 0.03 |
| | PixMix | 35.77 ± 0.01 | 33.84 ± 0.16 | 73.69 ± 0.28 | 62.37 ± 0.24 | 99.74 ± 0.01 | 99.39 ± 0.04 |

### E.2.3 VARYING REGULARIZATION METHODS

Table 9: Evaluation of robustness over different regularizations on CIFAR-100 benchmark with DenseNet backbone.

| Regularization | Values | mCE ↓ | $L_2$ (PGD) | $L_\infty$ (PGD) ↓ | Clean Acc. ↑ |
|---|---|---|---|---|---|
| Learning Rate | 3e-03 | 53.38 ± 0.53 | 99.64 ± 0.10 | 99.99 ± 0.02 | 60.84 ± 0.31 |
| | 1e-02 | 46.99 ± 0.20 | 99.41 ± 0.10 | 99.99 ± 0.01 | 70.59 ± 1.82 |
| | 3e-02 | 43.81 ± 0.71 | 99.51 ± 0.10 | 99.99 ± 0.01 | 72.55 ± 0.52 |
| Weight Decay($L_2$) | 0.0 | 44.98 ± 0.24 | 89.60 ± 0.40 | 99.83 ± 0.02 | 70.07 ± 0.11 |
| | 1e-4 | 43.49 ± 0.20 | 98.96 ± 0.08 | 99.99 ± 0.01 | 72.85 ± 0.35 |
| $L_1$ | 5e-7 | 42.51 ± 0.12 | 99.48 ± 0.09 | 99.99 ± 0.01 | 74.35 ± 0.10 |
| | 2e-6 | 43.35 ± 0.07 | 99.99 ± 0.01 | 99.57 ± 0.13 | 73.91 ± 0.15 |
| | 5e-6 | 43.23 ± 0.30 | 99.63 ± 0.07 | 99.99 ± 0.01 | 73.61 ± 0.22 |
| SPIDER (w/ best config) | - | 36.90 | 81.80 | 99.95 | 70.30 |

Effective regularization techniques significantly enhance the robustness of models. The application of SPIDER, in conjunction with optimal regularization methods, notably amplifies both common corruption and adversarial robustness.

