\le s(d_i) \le 1$, $0 \le \dfrac{c \cdot s(d_i) - 1}{c - 1} \le 1$, i.e. $0 \le s(d_i) - r_i \le 1$.

For the sake of simplicity, let $\gamma_i(d_i) := s(d_i) - r_i$. $r_i(d_i)$ is a monotonically decreasing function with respect to $d_i$ ($\because s(\cdot)$ and $c \cdot s(\cdot) - 1$ are monotonically decreasing functions.) Trivially, $0 \le \gamma_j(d_j) \le 1$ and $\gamma_j(\cdot)$ is a monotonically decreasing function.

Recap that $g_{ij}(x(t))^* = \dfrac{p_\eta(t) - p_\eta(1-t)}{p_\eta(t) + p_\eta(1-t)}$. Let us rewrite $g_{ij}(x(t))^*_{LS}$ as:

$$g_{ij}(x(t))^*_{LS} = \frac{\gamma_i(d_i)p_\eta(t) - \gamma_j(d_j)p_\eta(1-t)}{p_\eta(t) + p_\eta(1-t)}$$

One important thing to note here is that both $g_{ij}(x(t))^*$ and $g_{ij}(x(t))^*_{LS}$ are monotonically decreasing functions with respect to $t$ that are symmetric to the point $(1/2, 0)$.

**Lemma 1.** $g_{ij}(x(t))^*$ and $g_{ij}(x(t))^*_{LS}$ are monotonically decreasing functions with respect to $t \in [0, 1]$ that are symmetric to the point $(1/2, 0)$.

Another thing to note is that if we show $0 \le g_{ij}(x(t))^*_{LS} \le g_{ij}(x(t))^*$ for $t \in [0, 1/2]$ and $0 \ge g_{ij}(x(t))^*_{LS} \ge g_{ij}(x(t))^*$ for $t \in [1/2, 1]$, at the same time we are showing $\forall -1 \le \alpha \le 0 \le \beta \le 1, \Theta(f^*_{LS}, \alpha, \beta, x_i, x_j) \ge \Theta(f^*, \alpha, \beta, x_i, x_j)$ considering Lemma 1 (Figure 3a.)

**Lemma 2.** $0 \le g_{ij}(x(t))^*_{LS} \le g_{ij}(x(t))^*$ for $t \in [0, 1/2]$ and $0 \ge g_{ij}(x(t))^*_{LS} \ge g_{ij}(x(t))^*$ for $t \in [1/2, 1]$.

With Lemma 1 and Lemma 2, we can now finally derive that the boundary thickness of $f^*_{LS}$ is greater than or equal to $f^*$.

Let $0 \le t^{LS}_\beta \le 1/2 \le t^{LS}_\alpha \le 1$ be the values of $t$ such that

$$(t^{LS}_\alpha, t^{LS}_\beta) = \left( \inf_t g_{ij}(x(t))^*_{LS} = \alpha, \sup_t g_{ij}(x(t))^*_{LS} = \beta \right).$$

Since $g_{ij}(x(t))^*_{LS}$ is monotonically decreasing, the boundary thickness of $f^*_{LS}$ is $\|x_i - x_j\| \int_0^1 I\{\alpha < g_{ij}(x(t))^*_{LS} < \beta\}dt = \|x_i - x_j\| \cdot (t^{LS}_\alpha - t^{LS}_\beta)$.

Similarly, let $0 \le t_\beta \le 1/2 \le t_\alpha \le 1$ be the values of $t$ such that

$$(t_\alpha, t_\beta) = \left( \inf_t g_{ij}(x(t))^* = \alpha, \sup_t g_{ij}(x(t))^*_{LS} = \beta \right).$$

The boundary thickness of $f^*$ is $\|x_i - x_j\| \cdot (t_\alpha - t_\beta)$.

We now show that $t^{LS}_\beta \le t_\beta$ using Lemma 1 and Lemma 2, with the definition of $t^{LS}_\beta$ and $t_\beta$. For the sake of simplicity, let $h^*(t)$ and $h^*_{LS}(t)$ denote $g_{ij}(x(t))^*$ and $g_{ij}(x(t))^*_{LS}$ respectively.

$$h_{LS}^*(t_\beta^{LS}) = \beta \leq h^*(t_\beta^{LS}) \qquad \text{Def. of } t_\beta^{LS}, \text{Lemma 2}$$

$$h^*(t_\beta) = \beta \geq h_{LS}^*(t_\beta) \qquad \text{Def. of } t_\beta, \text{Lemma 2}$$

$$h_{LS}^*(t_\beta^{LS}) \geq h_{LS}^*(t_\beta) \qquad h_{LS}^*(t_\beta^{LS}) = \beta, \beta \geq h_{LS}^*(t_\beta)$$

$$t_\beta^{LS} \leq t_\beta \qquad \text{Lemma 1}$$

Likewise, we can trivially derive $t_\alpha^{LS} \geq t_\alpha$. Then, $(t_\alpha^{LS} - t_\beta^{LS}) \geq (t_\alpha - t_\beta)$, i.e.

$$\|x_i - x_j\|(t_\alpha^{LS} - t_\beta^{LS}) \geq \|x_i - x_j\|(t_\alpha - t_\beta). \qquad \square$$

## B.2 PROOF ON LEMMA 1

**Lemma 1.** $g_{ij}(x(t))^*$ and $g_{ij}(x(t))^*_{LS}$ are monotonically decreasing functions with respect to $t \in [0, 1]$ that are symmetric to the point $(1/2, 0)$.

*Proof.* We divide the proof into two parts. First we prove that $g_{ij}(x(t))^*$ and $g_{ij}(x(t))^*_{LS}$ are symmetric to the point $\left(\frac{1}{2}, 0\right)$, and then prove the functions are monotonically decreasing.

A function $f$ being symmetric to a point $\left(\frac{1}{2}, 0\right)$ indicates that $f\left(\frac{1}{2} + x\right) = -f\left(\frac{1}{2} - x\right)$.

$$g_{ij}(x(t))^* = \frac{p_\eta(t) - p_\eta(1-t)}{p_\eta(t) + p_\eta(1-t)}$$

$$g_{ij}\left(x\left(\tfrac{1}{2} + t\right)\right)^* = \frac{p_\eta(\frac{1}{2}+t) - p_\eta(\frac{1}{2}-t)}{p_\eta(\frac{1}{2}+t) + p_\eta(\frac{1}{2}-t)} = -\frac{p_\eta(\frac{1}{2}-t) - p_\eta(\frac{1}{2}+t)}{p_\eta(\frac{1}{2}-t) + p_\eta(\frac{1}{2}+

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

| | 1e-02 | $46.99 \pm 0.20$ | $99.41 \pm 0.10$ | $99.99 \pm 0.01$ | $70.59 \pm 1.82$ |
| | 3e-02 | $43.81 \pm 0.71$ | $99.51 \pm 0.10$ | $99.99 \pm 0.01$ | $72.55 \pm 0.52$ |
| Weight Decay($L_2$) | 0.0 | $44.98 \pm 0.24$ | $89.60 \pm 0.40$ | $99.83 \pm 0.02$ | $70.07 \pm 0.11$ |
| | 1e-4 | $43.49 \pm 0.20$ | $98.96 \pm 0.08$ | $99.99 \pm 0.01$ | $72.85 \pm 0.35$ |
| $L_1$ | 5e-7 | $42.51 \pm 0.12$ | $99.48 \pm 0.09$ | $99.99 \pm 0.01$ | $74.35 \pm 0.10$ |
| | 2e-6 | $43.35 \pm 0.07$ | $99.99 \pm 0.01$ | $99.57 \pm 0.13$ | $73.91 \pm 0.15$ |
| | 5e-6 | $43.23 \pm 0.30$ | $99.63 \pm 0.07$ | $99.99 \pm 0.01$ | $73.61 \pm 0.22$ |
| SPIDER (w/ best config) | - | 36.90 | 81.80 | 99.95 | 84.65 |

Effective regularization techniques significantly enhance the robustness of models. The application of SPIDER, in conjunction with optimal regularization methods, notably amplifies this robustness across all measured metrics.

Additionally, it is worth noting that SPIDER, when optimally configured, not only strengthens the model's robustness against adversarial attacks but also tends to improve overall model accuracy. This dual benefit underscores SPIDER's effectiveness in balancing robustness with high performance in various challenging scenarios.