# OpenReview forum: "Rethinking Label Smoothing as a Tool for Embedding Perturbation Uncertainty"
_ICLR.cc/2024/Conference — Submitted to ICLR 2024_

### Official Review · Reviewer_4KU5 · 2023-10-27

**Soundness:** 3 good
**Presentation:** 3 good
**Contribution:** 3 good
**Rating:** 6
**Confidence:** 3

**Summary:**

Intuitively, as data augmentation introduces more data, the annotations for this additional data must be adapted beyond the one-hot label. The author utilizes label smoothing to produce labels for the augmented data, specifically estimating confidence levels based on perturbation levels. To assess the effectiveness of the proposed technique, the authors carry out both empirical and theoretical analyses. They show that the method results in an expanded decision boundary and flatter minima. Furthermore, empirical tests on four benchmarks reveal performance enhancements using the proposed method.

**Strengths:**

1. The studied problem is important and the proposed method is reasonable.
2. The author conducts both empirical analyses and theoretical analyses to demonstrate the effectiveness of the proposed method.
3. Ablation studies have been conducted to assess the contribution of different components.

**Weaknesses:**

My major concern for this study is the lack of a proper baseline in the experiment section.
Specifically, the authors mainly conduct experiments by applying SPIDER to four different settings (no aug. AugMix, DeepAug, PixMix) and make comparisons with the original setting.
As the proposed method is an improvement over label smoothing, I think it is necessary to make comparisons with the original label smoothing method.
For example, one reasonable baseline is to add the same constant label smoothing to all images in the original setting.

**Questions:**

I find the title to be a bit misleading. I feel the word "rethinking" implies the study sheds insights into the mechanism of label smoothing, while the study mainly focuses on adapting label smoothing to the augmentation-based methods. I would suggest the author change it to a different word.

I think the related work on estimating the label distribution/uncertainty is highly related to this study and would suggest the authors discuss this connection[1,2].

1. Towards Understanding Ensemble, Knowledge Distillation, and Self-Distillation in Deep Learning
2. Label Noise in Adversarial Training: A Novel Perspective to Study Robust Overfitting

---

> ### Author Response · Authors · 2023-11-22
> **Thank you for sharing your time on reviewing our paper.**
>
> Thank you for dedicating your time to reading our paper.
>
> **W. As the proposed method is an improvement over label smoothing, I think it is necessary to make comparisons with the original label smoothing method (i.e. standard label smoothing).**
>
> A. To elaborate further on our ablation studies, the term "STD LS" actually represents standard label smoothing as you have suggested. Since the removal of SPIDER’s perturbation function will nullify the effect of exponential smoothing function, we have instead conducted experiments with standard label smoothing. The ablation table's results indicate that standard label smoothing when used solely fails to increase general model robustness, resulting in no improvements (even inferior) in common corruption robustness. To further support our findings, we are willing to provide additional experiments on standard label smoothing in the near future.
>
> **Q. Can the authors include the discussion w.r.t. the suggested works which seem to be highly related?**
>
> A. We have detailed the comparative analysis between the works you suggested and our own in the following comment. Your suggestion of these pertinent works is greatly appreciated.
>
> - Towards Understanding Ensemble, Knowledge Distillation and Self-Distillation in Deep Learning
>
> The paper's main objective is to explore the enhancement of model performance through the ensemble of Deep Neural Networks (DNN) and the distillation of this ensemble knowledge into another DNN. This research diverges from ours as our interest lies in general model robustness, not in knowledge distillation or ensemble methods.
>
> - Label Noise in Adversarial Training: A Novel Perspective to Study Robust Overfitting
>
> This study argues that in adversarial training, inputs altered by adversarial means are often labeled too confidently, highlighting the phenomenon of double descent during this process. This phenomenon occurs when DNNs are trained with noisy labels. To address the gap between the assigned labels and the actual labels of these adversarially altered inputs, the authors propose softening the labels through temperature scaling and interpolating the perturbed and original inputs.
>
> While our research partially shares common ground in terms of using less confident labels for altered inputs, there are significant theoretical and technical differences. Theoretically, we have demonstrated that isotopic perturbations combined with label smoothing enhance overall model robustness analyzing with boundary thickness and smoothness. The referenced study, however, is focused on the inherent conflict between hard labels and the true underlying labels of adversarially perturbed inputs. Technically, where the cited paper reduces label discrepancies in adversarial training through temperature scaling and interpolation, our approach involves isotopic perturbations with varying degrees of label smoothing based on perturbation magnitude.

---

### Official Review · Reviewer_Gg61 · 2023-10-30

**Soundness:** 2 fair
**Presentation:** 3 good
**Contribution:** 2 fair
**Rating:** 5
**Confidence:** 3

**Summary:**

In this paper, a new form of label smoothing technique named SPIDER is introduced, which connects data perturbation strength with smoothing confidence of true labels. Authors measure improvements of SPIDER by reflection of changes in flat minia and boundary thickness, with theoretical proof and empirical results. Additionally, SPIDER effectively enhances performance on robustness benchmarks, e.g., common corruptions, and L2 adversarial robustness.

**Strengths:**

1. The proposed method, SPIDER, is concise and effective.
2. Authors provide detailed theoretical proof and validate theories with sufficient empirical experimental results. SPIDER effectively enhances robustness models without hurting the clean accuracy.

**Weaknesses:**

1. SPIDER can be considered a variant of LS. In the meantime, standard LS can be seen as a specific example of a monotonically decreasing function (yet without a clearly defined definition in the paper). Nevertheless, it remains unclear whether and why SPIDER outperforms standard LS. The results shown in Table 6 demonstrate that standard LS actually performs worse than the baseline in terms of mCE metrics, which raises my doubts about whether the thickness of boundaries is necessarily linked to robustness against common corruptions. Also,  there is a lack of comparisons between standard LS and SPIDER in Tables 1-5 and Figure 2.
2. SPIDER improves flat minia compared with ''No LS'',  yet not to a sufficiently large extent. The empirical evaluation in Figure 2 indicates that compared with SPIDER, Augmix achieves flatter minia.
3. Multiple previous works have explored strategies for improving flat minia and boundary thickness, e.g., noisy mixup[1], HAT[2]. However, authors do not include them as baselines to compare.

[1] Boundary thickness and robustness in learning models, Yang et al. NeurIPS 2020.
[2] Reducing Excessive Margin to Achieve a Better Accuracy vs. Robustness Trade-off, Rade et al. ICLR 2022.

**Questions:**

Please refer to weakness first, an additional question is listed below.
1. In Table 6,  ''No LS'' corresponds to adversarial training with noise bounded with L2 norm and outperforms the baseline. I am curious about the outcomes when perturbations are applied while employing standard label smoothing, which more rigorously demonstrates the gap between standard LS and SPIDER

---

> ### Author Response · Authors · 2023-11-22
> **Thank you for sharing your time on reviewing our paper (1/2)**
>
> In response to your queries, we have carefully crafted our rebuttal to ensure clarity and respect towards your valuable insights.
>
> ---
>
> **W1-1. Why does SPIDER instantiation outperform standard LS in terms of mCE metrics? Since standard LS is a marginal case of monotonically decreasing smoothing function, doesn’t the results violate the essence of theorem 1?**
>
> A. You are correct in noting that standard label smoothing (LS) is a form of monotonically decreasing smoothing function. Our research into SPIDER highlights that when such smoothing functions are used ‘together’ with isotropic perturbation, they generally increase model robustness. Our ablation study was designed to explore if each element of SPIDER – the isotropic perturbation and the monotonically decreasing smoothing function – could individually match the overall performance of SPIDER. The results revealed that these individual components either provided limited or only partial improvements in robustness.
>
> **W1-2. There are lack of comparisons between standard LS and SPIDER in Tables 1-5 and Figure 2.**
>
> A. Regarding the comparison between standard LS and SPIDER in our experiments, we believed that a direct comparison with only standard LS was not essential for the aspects explored. In Tables 1 & 2, which focus on boundary thickness, our theorem suggests that any monotonically decreasing label smoothing can enhance boundary thickness in conjunction with isotropic perturbation functions. Hence, there was no implication that solely using standard smoothing methods would increase boundary thickness. Similarly, for Figure 2, our theorems about flatness only indicate that any $L_2$-norm bounded perturbation promotes the discovery of flatter minima. For our main experiments in Tables 3-5, we have partially shown that standalone standard smoothing does not provide overall model robustness, as indicated in our CIFAR-100 ablation experiments (Table 6).
>
> **W2. SPIDER improves flat minima compared with ''No LS'', yet not to a sufficiently large extent. The empirical evaluation in Figure 2 indicates that compared with SPIDER, AugMix achieves flatter minima.**
>
> A. In response to the concerns about the outlier case where AugMix outperforms AugMix + SPIDER, it is essential to consider the broader context of these findings. While SPIDER generally promotes flatter minima, as shown in our results and illustrated in Figure 2, there are nuances in its interaction with AugMix. In the CIFAR-10 experiments, AugMix achieves notably flat minima on its own, indicating a strong baseline for model generalization through diverse augmentation methods.
>
> In the specific case of 'AugMix + SPIDER', although there is a slight deterioration in flatness compared to AugMix alone, this does not necessarily imply inferior overall performance. The combination 'AugMix + SPIDER' still performs better than other baseline scenarios. For example, 'AugMix+SPIDER' shows a log loss value of around 2.5 at radius 50, which is more favorable compared to 'Naive+SPIDER' at around 3.0, 'PixMix+SPIDER' at around 2.5, and 'DeepAugment+SPIDER' at around 7.0. This indicates that while AugMix's flatness may be slightly reduced when combined with SPIDER, the overall performance remains robust.
>
> Furthermore, SPIDER enhances AugMix's performance in terms of boundary thickness, corruption, and adversarial robustness on CIFAR-10/100, and Tiny-ImageNet, as detailed in Table 3. This suggests that the slight reduction in flatness is a trade-off for improvements in other critical aspects of model robustness. It is important to recognize that the interplay between flatness and other performance metrics is complex and not fully understood yet. Therefore, while the specific case of 'AugMix + SPIDER' may seem like a strong outlier in terms of flatness, it represents a strategic balance, enhancing certain aspects of robustness at the cost of a minimal decrease in flatness. This nuanced understanding is crucial in evaluating the overall effectiveness of model enhancements and underscores the challenges in conducting a systematic analysis in such dynamic and multifaceted scenarios.

---

> ### Author Response · Authors · 2023-11-22
> **Thank you for sharing your time on reviewing our paper (2/2)**
>
> **W3. Diverse methods aimed to improve flat minima and boundary thickness such as noisy mixup and HAT can be included as baselines.**
>
> A. We acknowledge your suggestion and agree that incorporating methods like noisy mixup and HAT could have added value to our study. However, there were specific reasons for their exclusion. Firstly, although mixup-based methods are theorized to enhance both common corruption and adversarial robustness, practical challenges such as the manifold intrusion issue, as identified in the AugMix paper, often lead to less effective results. This has been evidenced in various studies including those in AugMix and our own findings (refer to AugMix’s Table 1, Yang’s Figure 2b, and our Table 6). Therefore, we opted for previously established augmentation methods known for their significant impact on model robustness. Secondly, our research indicates that methods focusing exclusively on adversarial robustness tend not to improve, and may even detract from, robustness against common corruptions. Our primary objective was to formulate an augmentation strategy that holistically enhances model robustness, covering both common corruption and adversarial scenarios. Nevertheless, exploring SPIDER's performance in combination with these methods is an interesting prospect we acknowledge as valuable.
>
> **Q1. ''No LS'' corresponds to adversarial training with noise bounded with L2 norm and outperforms the baseline. What will be the outcomes when perturbations are applied while employing standard label smoothing?**
>
> A. On the topic of the “No LS” condition in our ablation study, it refers to models trained exclusively with isotropic perturbations, which are not adversarially generated. The combination of isotropic perturbations and standard label smoothing, as you've mentioned, constitutes another specific form of SPIDER. Thus, the main question to consider is whether this combination, another instantiation of SPIDER, contributes to general model robustness. We have provided some initial results of this SPIDER variant, as shown in the tables below, to illustrate its impact across various metrics and augmentation methods.
>
> | CIFAR10  | mCE original | mCE prtb + std LS | $L_2$ original | $L_2$ prtb + std LS | $L_\infty$ original | $L_\infty$ prtb + std LS |
> |----------|--------------|-------------------|-------------|------------------|-------------|-------------|
> | no aug.  | 25.57        | 14.02             | 68.57       | 21.50            | 99.52       | 48.85       |
> | AugMix   | 11.83        | 9.67              | 36.91       | 26.07            | 93.15       | 63.72       |
> | DeepAug. | 13.21        | 10.26             | 54.86       | 22.73            | 95.86       | 45.93       |
> | PixMix   | 11.86        | 9.05              | 65.37       | 31.06            | 99.69       | 79.53       |
>
>
> ---
>
> [1] Benchmarking Neural Network Robustness to Common Corruptions and Perturbations
>
> [2] MNIST-C: A Robustness Benchmark for Computer Vision

---

### Official Review · Reviewer_DLcS · 2023-10-31

**Soundness:** 2 fair
**Presentation:** 2 fair
**Contribution:** 3 good
**Rating:** 5
**Confidence:** 4

**Summary:**

This paper proposes "SPIDER", an adaptive label smoothing method with regard to the input perturbation intensity. More specifically, SPIDER perturbs input $x$ to $x + \delta$; the intensity of the target true label is defined as $s( \| \delta \|)$ where $s( \cdot )$ is a monotonically decreasing function. There are two theoretical results for the proposed method
- SPIDER increases boundary thickness (Yang et al) -- for any monotonically decreasing smoothing function $s( \cdot )$. This result holds for an optimal solution of each optimization method (SPIDER vs. perturbation only). Note that this result does not include comparisons between SPIDER vs. vanilla label smoothing vs. the non-perturbed version of SPIDER. Also, this result is orthogonal to the choice of $s$ where SPIDER uses an exponential decay function.
- SPIDER encourages flat minima (defined as the worst case loss function following Cha et al) where the theorem holds for a linear classifier with softmax activation. As far as the reviewer understood, this theorem showed that the solution space with input perturbations is a subset of the "flat" solution space; Again, this theorem is invariant to the choice of $s$ and, even invariant to label smoothing.

Experimental results include that (1) empirical boundary thickness on CIFAR-10/100 for non-SPIDER methods, SPIDER methods and input perturbation only methods, (2) empirical flatness on CIFAR-10/100 for non-SPIDER methods and SPIDER methods and (3) CIFAR-10/100, Tiny-ImageNet and ImageNet results with corrupted images, PGD attack and clean accuracy.

**Strengths:**

As far as the reviewer understood, the theoretical contribution of this paper is non-trivial. Note that I did not have enough time to understand the whole theorem thoroughly (especially, I couldn't check whether any error is in the proof, as well as any non-trivial assumption is in the proof), but I just skimmed the main proof techniques, and statements. Although the theorems cannot explain the design choice of the proposed method (e.g., exponential $s$), I think the theorems could be helpful in understanding the effects of label smoothing on boundary thickness and the input perturbation for flatness.

**Weaknesses:**

Many important details are missing in the main text. Even worse, although information is provided in the Appendix, there is no pointer in the main text (even at the beginning of the Appendix). Therefore, just reading the main text only provides very limited information. While reading the paper, I frequently faced trouble to understand the text. For example, there is no definition of $c$ in the main text, where the definition is introduced in E.1.1. The main text also did not provide detailed information on which architecture is used for the experiments, as well as optimization hyperparameters. I also think that in terms of the presentation, this paper has a large room for improvement. The current version is very difficult to understand.

For the theoretical results, I think each theoretical result needs nontrivial effort, but the link between the proposed method and the theoretical results is unclear to me. First, none of the theorems can explain the design choice of $\delta$ and $s$ which looks very complicated and requires two hyperparameters. If the theorems cannot explain the design choice, there should be an extensive ablation study for the choice of $\delta$ and $s$. Second, the second theorem is orthogonal to label smoothing but only depends on input perturbations. Considering that the second theorem needs a very limited situation (linear classifier), I think that it will be great if the second theorem is more related to the proposed method. Also, the reviewer presumes that the theorems also hold when $s$ is a constant function, i.e., the vanilla label smoothing. Please check my question in the below section.

Similarly, I think this paper needs more extensive empirical justification. For example, Theorem 1 shows that a scenario with perturbed input with monotonically decreasing label smoothing satisfies better thickness than vanilla training, and Theorem 2 shows that the solution space with the input perturbation is a subset of the "flat" solution space. These theorems do not show that the proposed method is an optimal method for solving each task (thickness, flatness), or better methods than other methods. Hence, I think this paper should compare the proposed method with other regularization methods. I doubt that increasing boundary thickness or flatness is a special property of the proposed method. For example, Yang et al. showed that L1 regularization, L2 regularization, large learning rate, early stopping, and cutout increase boundary thickness empirically. It may imply that a proper regularization method tends to increase boundary thickness. It means that, although this paper shows some theoretical results, it does not guarantee that the proposed method is actually a better method than other regularization methods.

In terms of the empirical contribution, the proposed method always needs input perturbations. Although adding an input perturbation is helpful in terms of theoretical properties, I think it limits the usage of the method to the recent state-of-the-art methods. For example, recent SOTA methods employ many strong augmentations, such as RandAug, many strong image view manipulations or mixed data sample augmentations (e.g., Mixup, CutMix). It is unclear whether the proposed method and the recent strong augmentations can work at the same time.

Finally, I think that the experimental results are based on very few baselines and weak backbones. A weak network has a large room for improvement, which could be a just issue of optimization. For example, as shown in "ResNet strikes back: An improved training procedure in timm", a different optimization setting invariant to other improved techniques can boost the performance by about +4%p for ResNet-50. Similarly, in the ImageNet classification field, it is important to show the generalizability of the method for various backbones, rather than a specific backbone. In terms of the comparison methods, the method should be compared with other methods targeted to robustness, such as Mixup, CutMix and ANT. There can be more related works to the generalizability, but I think this method should be compared to data augmentation methods and many label smoothing variants rather than the proposed design choice.

**Questions:**

Please check my comments on weakness. I have additional questions:

- I wonder if theorem 1 holds when $s$ is a monotonically non-decreasing function as well, for example, a constant function. It will make $\gamma_j$ is a constant function. From my gut feeling, Theorem 1 also holds for a constant function because a constant function is also a monotonic function, and the proofs only use the property of a monotonic function. If this is true, it means that Theorem 1 is also true for input perturbation + vanilla label smoothing; it will weaken the justification of the adaptive label smoothing because Theorem 2 is irrelevant to the adaptive label smoothing.
- Thm 4 subscription seems to be a typo. Maybe it is not $\cup_{i\in [n]}$, but $\cup_{R_i\in [n]}$, considering the Appendix.

Overall, I think this paper needs a non-trivial revision in the presentation (to make the submission clearer and self-contained), in empirical supports for each theorem, in the link between the theorem and the method, and the empirical contribution of the method.

---

> ### Author Response · Authors · 2023-11-22
> **Thank you for sharing your time on reviewing our paper (1/4)**
>
> We appreciate your feedback on our research. Given a number of issues highlighted in your review, we aim to address them to the best of our abilities. Our response might be somewhat lengthy as a result, but we would be grateful if you could dedicate time to read our detailed rebuttal thoroughly and respond, helping us uphold the high standards of ICLR together.
>
> ---
>
> - **No boundary thickness experiments on vanilla label smoothing or non-perturbed version of SPIDER.**
>
>   A. We focused on demonstrating SPIDER’s label smoothing component combined with isotropic perturbation in Tables 1 and 2, which is why standard label smoothing experiments or a non-perturbed version of SPIDER were not included. In addition, the non-perturbed version of SPIDER nullifies the effect of data augmentation ($s(0)=1.$), and therefore the results will be the same as reported values of 'original' columns. Simply put, the experiments were solely designed to support Theorem 1.
>
> - **Important details are lacked without pointers. e.g. $c$ in the main text. / architecture / optimization hyperparameters.**
>
>    A. We apologize if any part of our paper was lacking information causing readers hard to follow. On the other hand, we will try to advocate ourselves against the specific examples you have given. The architecture and optimization hyperparameters for main table is in Appendix E with pointer provided (“For additional details, refer to Appendix E.”) Despite that we have not explicitly mentioned $c$ as the number of classes, it could be inferred from Algorithm 1’s formulazations $\tilde{y} = [\tilde{y_1}, \tilde{y_2}, ..., \tilde{y_c}]$ and $\tilde{y_{i \in [c]}}$. However, we agree that a direct explanation of $c$ could enhance clarity and will add this, and have included the descriptions of $c$ in our Algorithm 1 as your suggestion (please see page 3 of the revised manuscript).
>
> - **The link between the proposed method and the theoretical results seems unclear.**
>
>   * *i) Theorems on boundary thickness and flatness do not seem to justify the specific design choices for perturbation and smoothing functions.*
>
>     A. Our specific model, i.e., SPIDER, falls into one of instantiations of our theorems regarding boundary thickness and flatness. Precisely, the exact functions used by SPIDER, i.e., $p_\gamma(\|z\|)=\text{Uniform}(0, \tau)$ and $s(z) := \text{clip}(e^{-\lambda(z-\epsilon)}, 0, 1)$ are one of the monotonically decreasing functions and perturbation function bounded by $L_2$ norm, respectively. Because the theoretical results are not restricted to a particular case and cover the broad range of functions of label smoothing and perturbations (including SPIDER), we truly believe that the theoretical advantages are also valid in the case of SPIDER. Therefore, it is hard to say that the link between the theorems and SPIDER is unclear.
>
>     When focusing on the theoretical reasons for the choices of smoothing and perturbations, we agree that we have not fully explored what the optimal function shape is regarding common corruption and adversarial robustness, as aforementioned in Section 5. While we have not exhaustively examined the optimal function shape, we have conducted minimal experiments with several smoothing functions, including standard smoothing, linear smoothing, and exponential smoothing functions. We found that the exponential smoothing function to be slightly superior in common corruption benchmarks.
>
>   * *ii) Theorems on flatness does not rigorously guarantee SPIDER’s ability to find flatter minima.*
>
>     A. That is correct. Albeit we have not directly formalized the SPIDER’s ability to find flatter minima on general DNNs or MLPs, we have assumed that SPIDER will indeed encourage a model to find flatter minima – which turned out to be generally true in our experimental cases (7 out of 8.) It is obviously challenging to build a theoretical statement covering all possible deep neural architectures, but we believe that the efforts to build a theory for the basic architecture, such as a single-layer perceptron, can be a valuable step for anticipating the behavior of deep models.

---

> ### Author Response · Authors · 2023-11-22
> **Thank you for sharing your time on reviewing our paper (2/4)**
>
> Also, as you pointed out, the second theorem only depends on the perturbations but is invariant to the smoothing. When clarifying the reason, the theorem translates the perturbed input to the perturbed parameters but does not connect the smoothed label of the perturbed inputs to the smoothed label of the perturbed parameters. Yet, we remain the label smoothing part on the level of conjecture. Therefore, the remaining job is to add the part of label smoothing to fully justify that the smoothed label on the input side directly links to the smoothed label, which means flatter loss, on the parameters side. We have provided the additional proposition and corollary to rigorously formalize that SPIDER indeed promotes flatter minima. The additional statements are named **Corollary 1** and **Proposition 1**, and we have added them in Appendix D of our revised manuscript (please see the part following the proofs on Theorem 3 and 4. on page 26). If accepted, by considering their importance, we will integrate the contents into our final paper's flatness part in the main paper rather than in the Appendix. Briefly speaking, for the high-level understanding, **Corollary 1** is the final statement, and it directly connects the perturbation on the input side to the flatness of the parameter side, i.e., (when borrowing the notation in the revised manuscript) $\theta_{\gamma}^{*}$, which is trained by perturbation-based augmentation, yields **the flatter behavior of loss surface on the parameter space.**
>
>
> - **Extensive empirical justification required on theorems.**
>
>   * *i) No theorems are guaranteeing the optimal performance w.r.t. boundary thickness and flat minima metrics.*
>
>     A. You are indeed correct. However, we would like to gently point out that the challenge you mention is an ongoing one in the field of deep learning. To our knowledge, no works have given optimal solutions for finding flat minima, both theoretically and empirically. Regarding boundary thickness, while the authors of the work [4] have made assumptions to show that mixup is optimal solution to increase boundary thickness, the assumptions made are unrealistic for real-world applications (Proposition A.1.) Furthermore, experimental results in ours (Table 6) and the authors’ (Figure 2) hints us that mixup may provide large room for boundary thickness but is inferior to our baseline augmentations in robustness benchmarks. Striking the balance between theoretical and empirical results seems to be the crucial aspect in this machine learning field.
>
>   * *ii) Theorems do not illustrate SPIDER’s superiority over other baseline methods or existing regularization methods (e.g. L1/L2 regularization, large learning rate, early stopping).*
>
>     A. Without intending any provocation, our primary aim was to show SPIDER's applicability in various situations to improve model robustness, rather than proving its superiority over existing methods. However, for comparative purposes, we are currently training the same backbone network (DenseNet) deployed in the original boundary thickness paper and will release the results within 12 hours. (Note: We have updated additional experimental section E.2.3 with results comparing SPIDER against existing regularization methods. Please take a moment to review these new insights.)
>
> - **Since SPIDER is input perturbation based algorithm, it may be hard to be combined with other augmentation methods such as RandAug, Mixup, and CutMix.**
>
>   A. Usage of SPIDER with different augmentations is possible and is indeed desirable to further boost model robustness, as demonstrated in our experiments. When applying SPIDER, we have regarded the augmented image distribution as the true distribution.  Specifically, on top of the augmented images, we have added isotropic perturbations with corresponding label smoothings, i.e., SPIDER. When pointing out the case done in our work, PixMix handles two different images to combine a single augmented image, which follows a similar mixing process to Mixup and CutMix, and we successfully combine SPIDER with the paired-image-based augmentation. PixMix is applied to augment the original images and our method is applied on the augmented dataset.
>
>   All the experimental results in the main table have been obtained with such a training scheme. We strongly believe that our method can be easily combined with other augmentations.

---

> ### Author Response · Authors · 2023-11-22
> **Thank you for sharing your time on reviewing our paper (3/4)**
>
> - **Weak backbone networks are exploited. A weak network has a large room for improvement, which could be a just issue of optimization.**
>
>   A. Paraphrasing your concern, your question is that the reported values may not be optimal for each of the baselines, and SPIDER’s robustness gain may be a result of a certain optimization method. Respectfully, we have adhered to standard backbone models (Wide-ResNet, ResNet) and optimizations (SGD momentum with cosine learning rate decay) used in notable works (PixMix and AugMix). These are the most popular networks that have played as backbones of the immense number of deep learning algorithms, due to their considerable performance and the wide range of applicability. Also, we want to emphasize that while optimizing these methods might further improve robustness, all methods, including ours, are evaluated under the same conditions, ensuring no advantage in optimization techniques. At this moment, to the best of our understanding, it is hard to imagine an unexpected factor that probably leads us to misunderstand somewhat unintended gains as the true gains from SPIDER.
>
> - **Diverse backbone networks should be deployed to demonstrate the generalizability of SPIDER.**
>
>   A. We think it is a meaningful question to inquire about. It has been demonstrated in AugMix (Table 1) and DeepAugment (Table 4) that while precise statistics may vary, the tendency of augmentations on improving model robustness have not changed among varying model architectures. Echoing findings from the prior works, we are currently running SPIDER’s performance over different backbone architectures and updating the results on Appendix E.2 (please see pages 29-30 of the revised paper). Our method shows consistent gains over different backbones. Kindly note that due to our limited computational resources and limited time budget, we are not able to conduct experiments on ImageNet experiments with different backbone networks.
>
> - **Comparisons with augmentation methods like Mixup, CutMix, or ANT are required.**
>
>   A. Due to computational limits, we were unable to test all augmentations mentioned in our Related Work. We chose specific strategies to demonstrate SPIDER's effectiveness. Our decision to exclude Mixup, CutMix, and ANT is explained in more detail: Mixup and CutMix were less effective in the AugMix study (Table 1), and ANT required additional training for noise generators, leading to unstable learning curves, lengthy training, and less effectiveness compared to AugMix.
>
> **Q1. Does theorem 1 hold for vanilla label smoothing + isotropic perturbation?**
>
> A. Yes. It seems your question has two parts: 1. Why we chose an exponentially decaying function over a standard label smoothing function, and 2. Whether the standard smoothing function's similar robustness improvements diminish the significance of our contribution.
>
> To answer the first part, we have chosen the exponentially decaying function simply because it marginally outperformed the standard label smoothing function in CIFAR common corruption benchmarks. In the following table, "exponential" indicates the SPIDER instantiation in the main paper, and "standard" indicates the version of SPIDER with standard label smoothing for function s(.).
>
> | Benchmark | s(.)        | No Aug. | AugMix | DeepAug. | PixMix |
> |-----------------|-------------|---------|--------|----------|--------|
> | CIFAR10-C |exponential|14.37|9.17|11.18|8.35|
> ||standard|14.02|9.67|11.26| 9.05|
> |CIFAR100-C| exponential |38.62|33.56|34.64|30.98|
> ||standard|39.07|34.45|34.60| 32.51|
>
> For the second part, we would like to emphasize our key contribution is answering “how label smoothing should be combined with perturbations to broadly enhance model robustness,” not on exponentially decaying function itself. This nuance is conveyed through the titles of Algorithms 1 and 2. Precisely, we have demonstrated that isotropic perturbations encourage flatter minima, and any monotonically decreasing smoothing function w.r.t. the intensity of perturbations will increase boundary thickness, leading to improvements in model robustness.
>
> **Q2. Notation mistake in $\cup_{i\in[n]}$?**
>
> A. We have used $[n]$ as the index set notation indicating integers ranging from 1 to n. Therefore, $\cup_{i \in [n]} R_i$ can be interpreted as $R_1 \cup R_2 \cup \cdots R_n$.
>
> ---

---

> ### Author Response · Authors · 2023-11-22
> **Thank you for sharing your time on reviewing our paper (4/4)**
>
> If at any stage we have caused you annoyance with your review, we offer our apologies. Please understand that our intent was to diligently advocate for our perspective and clarify our stance on the points you raised.
>
> We truly respect your perspective on our research and would like to offer our final thoughts. Our works have limitations in the aspect that extensive experiments on large scale backbone networks and training dataset have not been carried out, and the optimal function shapes for both perturbations and smoothing functions remain underexplored. Nonetheless, our work has solid contributions in the sense that we have provided systematical analysis and theoretical insights on how label smoothing should be incorporated with perturbations, and successfully enhanced general model robustness in different circumstances without a single exception unlike previous adjacent studies which were often limited to partial robustness or lacked theoretical understandings.
>
> We understand the difficulty in reversing your initial assessment, as it might appear to compromise the integrity of your review. Nonetheless, adjusting your point of view is fine on any stage of the discussion and even illuminate your openness and transparency in participating in this process. If our rebuttal addresses much of your concerns, we would truly appreciate reevaluation of our work with an open mind. Should there be any unresolved queries, please reach out to us prior to the rebuttal deadline, enabling us to offer further clarifications or additional information as needed. Thank you for dedicating time to our rebuttal.
>
>
> Best Regards,
>
> Authors.
>
> ---
>
> [1] SWAD: Domain Generalization by Seeking Flat Minima
>
> [2] What Are Effective Labels for Augmented Data? Improving Calibration and Robustness with AutoLabel
>
> [3] Label Smoothing and Logit Squeezing: A Replacement for Adversarial Training?
>
> [4] Boundary thickness and robustness in learning models
>
> [5] Overcoming Catastrophic Forgetting in Incremental Few-Shot Learning by Finding Flat Minima

---

> ### Comment · Reviewer_DLcS · 2023-11-23
>
> Thanks for your hard work during the rebuttal period. I have checked the authors' response, the other reviewers' comments, and the revised paper carefully. My response is also lengthy, therefore I listed the summary here.
>
> 1. I still think this paper needs more empirical verifications on their design choice. For example, the choice of LS.
> 2. It is because the theorems cannot fully support SPIDER design choice. Each theorem only (partially) supports each component of SPIDER (e.g., LS part and input perturbation, where LS part is universal to any non-increasing function).
> 3. The additional experiments in Appendix E are not fully satisfied, but acceptable to me. Thanks for your efforts
> 4. Despite of **2**, the additional theoretical clarifications in D could be helpful for readers.
> 5. Considering the above bullets, I revised my score to 5 from 3 (the other scores were also updated). I still vote for 5, because of **1** and **2**.
>
> ----
>
> First, I would appreciate the authors for addressing my concerns during a short rebuttal period. I think the additional experiments in E.2.2 and E.2.3 are somewhat insufficient (e.g., more comparison methods are suggested by me, such as ANT, RandAug, Mixup, and CutMix, or the reviewers, such as noisy mixup, HAT), but I think it is an acceptable experiment. Also, the additional theoretical statements seem to be helpful for understanding the previous theorems. However, I think the titles of 3.2.2 and 3.2.3 still imply that the theorems fully support that SPIDER can achieve flatness and boundary thickness, while the theorems show the effect of label smoothing and input perturbations, respectively. Therefore, it is somewhat non-trivial if the theorems still hold even if both of them are used at the same time. I think it could be an okay contribution, but it should be clarified in the main text. I still think that it is impossible that the current main manuscript cannot assume information in the Appendix, but I believe that it could be revised with a minor modification.
>
> I think the responses for "i) No theorems are guaranteeing ...." and "ii) Theorems do not illustrate ...." are not fully satisfied, but I think it is acceptable. I think the discussions in (i) is better to be included in the revised paper (e.g., in the Appendix).
>
> I also think that the response for "Since SPIDER is input perturbation based algorithm, it may be hard to be combined with other augmentation methods such as RandAug, Mixup, and CutMix." needs more precise verification (I agree that SPIDER is applicable by just treating the augmented samples as an input). For example, showing the actually performance of Mixup + SPIDER. As far as the reviewer knows, there is no empirical study showing input perturbation + SOTA augmentation methods on corrupted accuracy and attack accuracy. Particularly, the authors argued that "Mixup and CutMix were less effective in the AugMix study", but this experiment is conducted without input perturbations. I think it could be helpful to include the experiments in future revision.
>
> It is still unclear why the current design choice is proposed. As clarified by the authors, the first theorem holds for general label smoothing (including constant one) and the second theorem is not for input perturbation + exponential label smoothing, but only for input perturbation. I think it is unclear whether the addition of methods satisfies the linearity of their theoretical properties. For example, Will (method A + method B) show the same theoretical behavior with (method A) and (method B), even if the theorems are proved in different environments? (e.g., different input scenario or different target label scenario). I am not arguing that this paper should prove an unified theorem, but therefore this paper needs more empirical studies on its design choice. Also, the response says that "our key contribution is answering “how label smoothing should be combined with perturbations to broadly enhance model robustness,” not on exponentially decaying function itself", but in this case, I think more careful studies on the design choice should be conducted. I think the additional experiment in the response (vanilla LS vs. exponential LS) could be an acceptable empirical verification, while I cannot find linear LS experiments corresponding to "we have conducted minimal experiments with several smoothing functions, including standard smoothing, linear smoothing, and exponential smoothing functions".
>
> I partially agree with the comment "No boundary thickness experiments on vanilla label smoothing or non-perturbed version of SPIDER", where the experiments are designed to support the theorems. However, as the theorems can support *any* label smoothing methods, and *any* input perturbation, I think we need empirical studies to show how the selected design choice helps boundary thickness.

---

> > ### Author Response · Authors · 2023-11-23
> >
> > Dear reviewer DLcS,
> >
> > Thank you for your detailed review and the time you invested in reevaluating our manuscript. Your insights and the subsequent adjustment of the evaluation score are greatly appreciated. We are pleased to note that our efforts in addressing your concerns have been recognized.
> >
> > Sincerely,
> >
> > Authors

---

### Official Review · Reviewer_M1mD · 2023-11-01

**Soundness:** 2 fair
**Presentation:** 4 excellent
**Contribution:** 3 good
**Rating:** 6
**Confidence:** 3

**Summary:**

This paper targets the problem of training robust neural networks on image datasets. To achieve this, the authors propose to use data augmentation for input images as well as label smoothing for augmented images. Specifically, a new perturbation function for augmentation sampling and a label smoothing function are proposed to augment input images and labels. Experiments on some image datasets show that compared with no augmentation and previous input data augmentations, the proposed method is capable of increasing robustness in terms of common corruptions and adversarial attacks.

**Strengths:**

(1) The paper is well written and easy to follow.

(2) This work presents two new techniques, a perturbation function for augmentation sampling and a label smoothing function. The proposed new technique is supported by the findings that smoothed labels can increase boundary thickness and input perturbations are related to parameter perturbations in the linear case, which can facilitate flatter minima through empirical verification.

**Weaknesses:**

(1) It is mentioned in the manuscript that robustness is important in terms of different kinds of distributional shift including domain shift. I understand that the main goal is to mitigate perturbations, but according to the description of the corruption types in the appendix, it seems that some corruptions are quite strong and humanly perceptible. Therefore, the robustness with respect to domain shift also worth studying but such experiments are missing. It is also mentioned in section 3.3.1 that flat minima, which can be achieved by the proposed method according to both theoretical and empirical verifications, can improve robustness with respect to domain shift, but this is not empirically verified for the proposed method.

(2) The proposed method can also improve robustness with respect to adversarial attacks. However, there is no comparison with other popular adversarial learning algorithms.

(3) If I’m understanding correctly, the main baselines used in the experiments are pure image-based augmentation strategies. Class labels are not used in these baselines for augmentation. In my opinion, it is possible to incorporate augmented labels for these baseline methods as well. For example, it could be labels with reduced confidence for the ground-truth class for augmented images. It seems that comparing with these non-label augmentations looks unfair for the proposed label-smoothing method as it can leverage more information. Apart from this, ablation results in Table 6 do not show universal improvements under each component.

Other questions and minor issues:

- In the empirical verification of the flatness of SPIDER, I’m not sure how good the models are in Figure 2. What are the final accuracies of these models and are they the same models used to report the performance in Table 3?
- I understand that the theoretical properties for input perturbations in the non-linear case is difficult to handle. The provided results in the linear case may not hold for deep neural networks and they are not rigorously connected with the real experiments.

**Questions:**

See weakness part.

---

> ### Author Response · Authors · 2023-11-22
> **Thank you for sharing your time on reviewing our paper (1/2)**
>
> We sincerely appreciate the time and effort you have dedicated to reviewing our paper. Your insightful comments and constructive feedback are invaluable to us, and we are grateful for the opportunity to address your queries and concerns.
>
> ---
>
> **Q. Domain shift may be a good benchmark for demonstrating SPIDER’s effectiveness.**
>
> A. Indeed, your observation is insightful. While our research has not yet explored SPIDER's performance against domain shift, we expect that SPIDER would likely improve robustness either as a standalone method or in conjunction with prior augmentation methods as in our experimental results on common corruption and adversarial robustness benchmarks. Despite we have attempted to conduct basic experiments using PACS dataset (which is one of the most widely used DG task benchmarks with four different domains), we were not able to conduct experiments successfully within the rebuttal period. Nonetheless, we still expect SPIDER to improve model robustness against domain shifts as well.
>
> **Q. Methods that directly aim for adversarial robustness is not included in the baseline methods.**
>
> A. Your point is well-taken. Including adversarial training (AT) methods in our experiments would be ideal. We value your suggestion and have run additional experimental results on CIFAR100 regarding adversarially trained model in the below table. AT indicates a typical adversarial training with adding adversarial noise to the clean image. By combining with the adversarial training (AT), SPIDER shows consistent gains and a remarkable gain for the $L_\infty$ case, i.e., over +20.33\% gains. If accepted, we will conduct full adversarial training experiments on the main table's benchmarks and report the statistics in our final paper as your suggestion.
>
> |     Metrics     |       mCE      |       |      $L_2$     |           |   $L_\infty$   |    |   clean acc.   |      |
> |:--------------:|:--------------:|:--------------:|:--------------:|:--------------:|:--------------:|:--------------:|:--------------:|:--------------:|
> |     Method     |    original    |     +SPIDER    |    original    |     +SPIDER    |    original    |     +SPIDER    |    original    |     +SPIDER    |
> | AT($L_\infty$) | 43.16$\pm$0.17 | 42.59$\pm$0.41 | 55.66$\pm$0.15 | 54.69$\pm$0.42 | 80.97$\pm$0.08 | 60.64$\pm$0.40 | 60.27$\pm$0.34 | 60.64$\pm$0.40 |
>
>
> **W3-1. Concerns about the fairness of evaluating SPIDER against baseline augmentations that do not manipulate labels.**
>
> A. We would like to respectfully address this concern by highlighting two key points. Firstly, our primary goal is not to demonstrate the superiority of one method over another. Rather, our focus is on emphasizing the effectiveness of SPIDER in enhancing the overall robustness of models, whether it is used alone or in tandem with other augmentation techniques in diverse circumstances. Moreover, the assertion that SPIDER has an unfair advantage because it alters class labels, unlike other augmentation methods, in some sense might oversimplify the situation. For instance, SPIDER operates without any inductive bias related to the dataset, whereas other baseline methods may have inherent advantages. AugMix, for example, extensively employs image operations known for their beneficial effects on image classification. Similarly, PixMix utilizes additional images beyond the training set, and DeepAugment relies on a pretrained image-to-image model which deploys additional image dataset and network. These aspects could be viewed as providing an edge to these methods when compared with SPIDER, which solely depends on perturbations and label smoothing. Therefore, we humbly suggest that there is no inherent unfair advantage in favor of any particular augmentation approach in our evaluations.
>
> **W3-2. Ablation results in Table 6 do not show universal improvements under each component.**
>
> A. Indeed, this observation is accurate. SPIDER’s two components, which are label smoothing and perturbations, do not show incremental gains when we add one component to the other. Nonetheless, it is pivotal to underscore the collective significance of SPIDER as an integrated system, rather than focusing on its individual components. When standard label smoothing is applied in isolation, it enhances adversarial robustness but fails to make a discernible impact on common corruption robustness. In contrast, the use of isotropic perturbations alone yields only mediocre improvements in robustness, particularly when compared to the comprehensive efficacy of SPIDER or other augmentation methodologies. Consequently, we advocate that the holistic application of SPIDER is critical for achieving substantial and broad-based enhancements in model robustness, steering clear of incremental or minor improvements.

---

> ### Author Response · Authors · 2023-11-22
> **Thank you for sharing your time on reviewing our paper (2/2)**
>
> **Q1. What are the final accuracies of the models used for flatness analysis in Figure 2? Are they the same models used in Table 3?**
>
> A. That is exactly correct. The models for the flatness analysis exactly show the final accuracies in Table 3.
>
> **Q2. The theoretical handling of input perturbations in non-linear scenarios is complex, and the connection between flatness theorems and practical experiments may not be robust.**
>
> A. We agree with your assessment, which is one reason there was an outlier in our flatness experiments. We maintain that SPIDER is designed to 'encourage' the discovery of flatter minima, acknowledging the intricacy of the theoretical-experimental relationship. It is obviously challenging to build a theoretical statement covering all possible deep neural architectures, but we believe that the efforts to build a theory for the basic architecture, such as a single-layer perceptron, can be a valuable step for anticipating the behavior of deep models. Based on our theory, we have found that SPIDER is indeed shown to encourage deep models to find flatter minima in evaluation.
>
> ---
>
> [1] Benchmarking Neural Network Robustness to Common Corruptions and Perturbations
> [2] MNIST-C: A Robustness Benchmark for Computer Vision

---

### Author Response · Authors · 2023-11-23
**Manuscript Update: Incorporating Your Suggestions and Enhancements**

In this document, we outline the updates made during the period of response to feedback:

- **Adjustments to the Title**

  The term 'Rethinking' has been switched out for 'Utilizing' in the title. (page 1)

- **Clarifications in the Algorithm**

  A detailed explanation of the variable $c$ has been added to Algorithm 1. (page 3)

- **Enhanced Theoretical Framework**

  A new proposition and corollary have been introduced at the last part of Appendix D, rigorously formalizing how perturbations can encourage flatter minima. (page 26)

- **Expanded Experiments with Different Architectures**

  We have conducted and included additional experiments using various model architectures, detailed in Appendix E.2.2. (pages 29-30)

- **Broader Experiments with Diverse Regularization Techniques**

  Further experiments using a range of regularization methods are now included in Appendix E.2.3. (page 30)

---

### Meta-Review · Area_Chair_Qwyi · 2023-12-11

**Metareview:**

In this paper the authors introduce a method to robustly train neural networks for image classification. Specifically, the authors propose employing label smoothing in tandem with data augmentation. The authors demonstrated the efficacy of their approach on image classification with CIFAR-10, CIFAR-100 and ImageNet and measured the robustness in terms of the boundary thickness and flat minima. The authors compared their results to other baselines based on data augmentation. The reviewers commented positively on the quality of presentation, a concise and effective method and the theoretical justification for the method.

The reviewers also commented negatively on the lack of comparisons to adversarial robustness methods, the need for more empirical measures of robustness such as domain shift and the lack of connection between the theory and the implementation.

There was extensive discussion in the rebuttal phase and while the reviewers raised their scores, some of the issues above still existed. From my perspective, I particularly appreciate the need for more empirical evaluation of robustness. Many previous methods have been proposed in the past to improve generalization, yet showed limited efficacy in real world datasets. I would implore the authors to consider using more empirical measurements to capture the empirical gains of this approach. Given that no reviewer is a strong champion of this paper and the unresolved concerns surfaced by the reviewers, this paper will not be accepted to this conference but the authors are encouraged to address all of these concerns, run additional experiments and consider resubmitting to a future conference.

**Justification For Why Not Higher Score:**

Limited connection of theory to experiments. More empirical results.

**Justification For Why Not Lower Score:**

N/A

---

### Decision · Program_Chairs · 2024-01-16

Reject